# Slow update of internal representations impedes synchronization in autism

Gal Vishne [1,6✉], Nori Jacoby[2,6], Tamar Malinovitch[3], Tamir Epstein[4], Or Frenkel[5] & Merav Ahissar [1,5✉]

Autism is a neurodevelopmental disorder characterized by impaired social skills, motor and perceptual atypicalities. These difficulties were explained within the Bayesian framework as either reflecting oversensitivity to prediction errors or – just the opposite – slow updating of such errors. To test these opposing theories, we administer paced finger-tapping, a synchronization task that requires use of recent sensory information for fast error-correction. We use computational modelling to disentangle the contributions of error-correction from that of noise in keeping temporal intervals, and in executing motor responses. To assess the specificity of tapping characteristics to autism, we compare performance to both neurotypical individuals and individuals with dyslexia. Only the autism group shows poor sensorimotor synchronization. Trial-by-trial modelling reveals typical noise levels in interval representations and motor responses. However, rate of error correction is reduced in autism, impeding synchronization ability. These results provide evidence for slow updating of internal representations in autism.

[1] Edmond and Lily Safra Center for Brain Sciences, Hebrew University, Jerusalem, Israel. [2] Computational Auditory Perception Group, Max Planck Institute for Empirical Aesthetics, Frankfurt, Germany. [3] Cognitive Science Department, Hebrew University, Jerusalem, Israel. [4] Psychiatric Division, Sheba Medical Center, Tel-Hashomer, Israel. [5] Psychology Department, Hebrew University, Jerusalem, Israel. [6]These authors contributed equally: Gal Vishne, Nori Jacoby. ✉email: gal.vishne@gmail.com; msmerava@gmail.com

The core difficulty in social interactions of individuals with ASD has traditionally been attributed to a lack of social interest and motivation[1], but this view has been recently challenged[2]. Recent studies revealed that atypical perceptual and motor processing are consistent characteristics of autistic experience[3]. Individuals with ASD show particular difficulties when sensorimotor integration is required[4,5], and their magnitude is correlated with symptom severity[6]. The manifestation of various sensory and sensorimotor atypicalities suggests that cross-modal accounts may be required to explain this complex phenotype within a unified framework. Accordingly, several recent studies have attempted to explain autism within the cross-modal Bayesian framework. This framework attributes difficulties to an abnormal estimation of the environment's statistics, which leads to impaired integration of past experiences for regulating ongoing behavior[7–11]. Yet, the nature of this abnormality has been disputed.

A dominant account suggests that individuals with autism overestimate the rate of changes in the statistics of the external environment[10,12], leading to an overestimation of the reliability of recent events compared with earlier ones. Consequently, recent events are overly represented in the formation of perceptual estimations and motor plans ("increased volatility" hypothesis). An opposing account ("slow updating" hypothesis) proposes that individuals with autism are able to estimate environmental statistics correctly, yet the rate at which internal priors are updated is slower than neurotypical. This account was proposed by Lieder et al.[11], who used computational modeling of two-tone frequency discrimination to show that participants' responses are biased by the tones in previous trials. Yet, the relative weight of recent and long-term contributions differs between individuals with autism and neurotypical individuals. Early trials influenced perceptual judgments similarly in both groups, but the influence of recent trials was reduced in the autism group. Therefore, while the statistics of earlier events are integrated well into predictions and actions, this accumulation takes longer, and recent events are underweighted. Both theories have clear predictions for broad contexts, yet in many cases, these predictions are opposed. In particular, when fast online updates are needed for adequate task performance, the "increased volatility" hypothesis predicts better performance in autism, whereas the "slow updating" account predicts impaired performance. Synchronization tasks require a fast update of internal representations and motor responses based on external cues and therefore provide an experimental platform for comparison between these opposing predictions.

Synchronization ability was reported to be impaired in autism, in both social and nonsocial contexts[13–15]. Studies with neurotypical populations found that synchronization is functionally related to the theory of mind[16,17] and to social behavior[18]. The rationale proposed for these observations is that synchronized actions promote a predictive mechanism trained to anticipate other's actions and intentions[19,20].

Paced finger tapping is a synchronization task in which participants are asked to align their tap to the beat of an external metronome. Perfect synchrony means perfect alignment between the participant's taps and the external metronome. Human performance is limited in two aspects. First, participants tap with a small negative asynchrony, which is perceived as synchronous (Fig. 1a). The (mean) magnitude of this asynchrony is influenced by many factors, peripheral and central, including the type of movement, type of feedback, and characteristics of the metronome sound[21–25]. Since the relative contribution of the peripheral and central sources is not known, we had no prediction for group differences regarding mean asynchrony. The second limitation on synchrony is variability around this mean. Though tapping variability is also affected by both central (such as

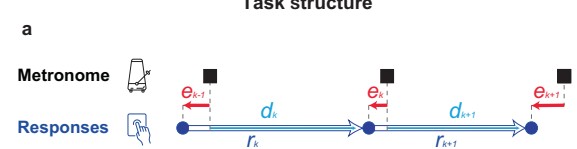

**Task structure**

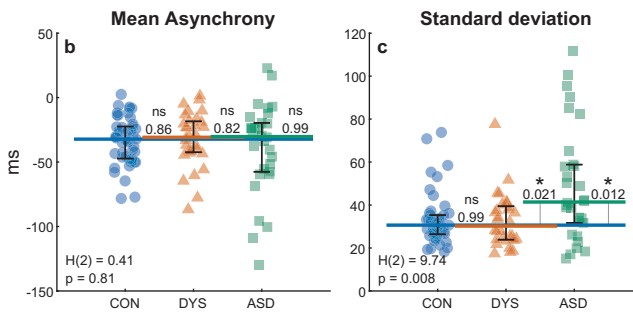

**Fig. 1 Isochronous finger tapping: mean asynchrony is similar in the three groups, but variability around this mean is substantially larger in Autism (ASD) compared to neurotypical (CON, control) and dyslexia groups (DYS). a** A schematic illustration of the temporal structure of paced tapping: metronome stimuli (presented every 500 ms, black squares), and finger-tap responses (blue circles) as a function of time; $e_k$ - error (asynchrony, typically negative) in tap $k$; $r_k$ - inter-tap interval; $d_k$ - delay interval from the previous metronome stimulus (beat $k$-1) to the following finger tap (tap $k$). Note that $r_k = d_k - e_{k-1}$. **b, c** Basic tapping parameters: **b** Mean asynchrony is negative for all three groups ($p < 0.001$) and similar in the three populations, though more broadly distributed in the ASD group. **c** Standard deviation is larger in the ASD group compared to the two other groups. Each dot represents the performance of one participant (average of two blocks); the y-axis represents the score in ms, and x-axis and color represent group membership (with a small jitter for readability): blue circles —neurotypical, red triangles—dyslexia, and green squares—ASD. The median of each group is denoted as a line of the same color; error bars around this median denote an interquartile range. Kruskal–Wallis $H$-statistic and corresponding $p$ values are plotted in the bottom-left corner; $p$ values of comparisons between groups are plotted next to the line connecting the groups' medians. $N = 109$ subjects ($N_{CON} = 47$, $N_{DYS} = 32$, $N_{ASD} = 30$). Source data are provided as a Source Data file. Though there are a few outlier results in both mean asynchrony and standard deviation of participants with ASD, these are not the same individuals—scores on these two measures were not correlated in the ASD and dyslexia groups (Spearman correlations: $\rho_{ASD} = -0.2$ ($p = 0.3$), $\rho_{DYS} = -0.24$ ($p = 0.18$)). A significant correlation was found only in the neurotypical group ($\rho_{CON} = -0.37$, $p = 0.01$, uncorrected). Statistical tests are two-sided unless stated otherwise.

intelligence[26,27]) and peripheral factors, the contribution of peripheral factors, such as motor noise, is considerably smaller[28]. Importantly, the components underlying variability were systematically modeled.

When the metronome tempo is constant, models of paced finger tapping assume that keeping the variability small is challenged by two sources of noise: noise in motor responses and noise in the internal representation of the metronome tempo (timekeeping). Both can be corrected online by using the asynchrony error signal (the perceived interval between the metronome beat and the tap). If errors are not corrected quickly and are kept through metronome beats, they accumulate, increasing the variability around the mean asynchrony and leading to poor synchronization[28–32]. Changing environments introduces another difficulty—to identify when and to what extent the metronome tempo changes and quickly correct for it. This is done

by modifying the internal representation of the external tempo, while concurrently correcting for the stationary noise sources mentioned above. The "slow updating" hypothesis predicts that the rate of error correction will be reduced in autism while motor noise and timekeeping noise will be similar to that of neurotypical individuals. In contrast, the "increased volatility" hypothesis predicts increased (over-)correction, leading to either superior alignment or even overshooting the amount of correction required.

To test the specificity of tapping atypicalities to autism, we also recruited a group of participants with dyslexia, matched for age and cognitive reasoning. Dyslexia, a common neurodevelopmental disorder, is characterized by poor reading and spelling[33]. Similar to individuals with ASD[34], individuals with dyslexia show high concurrence with ADHD[35], and atypical perceptual characteristics[36,37]. But individuals with dyslexia are not diagnosed for social difficulties.

In this work, we administer two tapping protocols, one using a fixed metronome tempo (Experiment 1), and the other using a tempo-switch protocol (Experiment 2). Together, the two experiments allow us to quantify the dynamics of error correction in both stationary and changing environments. For both experiments, we use computational modeling to quantify the rate at which internal representations are updated, and dissociate its contribution to task performance from that of internal noise sources. Only the autism group shows impaired synchronization, owing to reduced use of recent sensory information for error correction. Noise levels in both interval representation and motor execution are intact. These results support the "slow updating" account of autism.

## Results

### Experiment 1 – isochronous tapping reveals reduced online error correction in ASD, but not in dyslexia.
As a main measure of performance, we used asynchrony (the difference between metronome stimulus and participant responses). We measured the mean and standard deviation (SD) of the asynchrony in a paced finger-tapping task, with a fixed 2 Hz auditory metronome beat (illustrated in Fig. 1a; test-retest correlation of the main tapping parameters is ~0.8; Supplementary Note 1 and Supplementary Fig. 1).

We recruited three age and cognitive matched groups (Supplementary Table 1)—neurotypical individuals ($N_{CON} = 47$), individuals with dyslexia ($N_{DYS} = 32$), and individuals with ASD ($N_{ASD} = 30$). As expected, the mean asynchrony manifested by most participants was negative (105/109 participants: 96.3%). Mean asynchrony was similar in the three groups (average over two repetitions, median [interquartile range] (ms): neurotypical: −32.2 [24.9], dyslexia: −30.8 [24], autism: −30.3 [37.9], Kruskal–Wallis test $H(2) = 0.41$, $p > 0.8$, Fig. 1b).

By contrast, we found significant differences in the variability (denoted by the SD) of the groups around their mean asynchrony (average over two repetitions, median [interquartile range] (ms): neurotypical (CON): 30.6 [8.9], dyslexia (DYS): 30.2 [15.6], autism (ASD): 41.4 [27.1], Kruskal–Wallis test $H(2) = 9.74$, $p = 0.008$, see Fig. 1c). The significant group difference was due to the large variability of individuals with ASD (post hoc analysis of ASD group vs. neurotypical or dyslexia using two-sided Tukey–Kramer method (throughout the paper): $p < 0.022$, Cliff's delta > 0.38 in both cases), while there was no difference between the dyslexia group and the neurotypical group ($p > 0.95$). Although there were individuals with autism whose SD was in the range of the neurotypical population, the SD of a third of the group was more than two standard deviations (of the neurotypical distribution) above the neurotypical mean, compared with only one individual with dyslexia whose variability was in this range. This pattern of results was replicated in Experiment 2 (Supplementary Figs. 2 and 3).

*Reduced online error correction underlies poor synchronization in ASD.* Phase correction is the process of using the perceived error (deviation of the current tap from mean asynchrony) to adjust the timing of the next tap to be closer to the participant's mean asynchrony (which is perceived as synchronous with the metronome beat). To test the efficiency of online phase correction we calculated the correlation between consecutive asynchronies (errors). Any positive correlation means that errors tend to persist across beats, and a correlation of one means that errors are fully retained across consecutive beats. A correlation of zero means that errors were not carried across trials, and negative correlations mean overcorrection. All three groups showed a positive correlation (Fig. 2a–c, $r_{CON} = 0.60, r_{DYS} = 0.59, r_{ASD} = 0.75$),

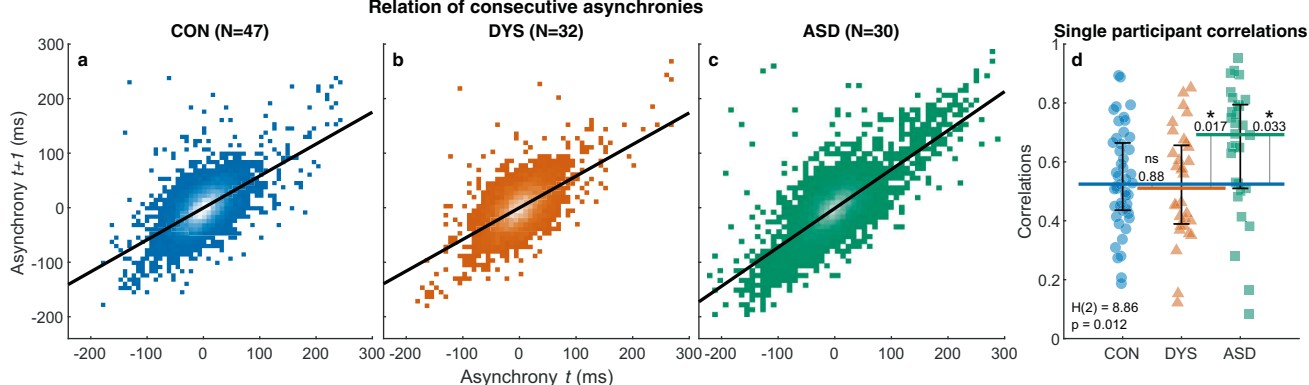

**Fig. 2 Correlation between consecutive asynchronies (errors) is highest in the ASD group revealing reduced online error correction. a–c** Scatter plots showing correlations between consecutive asynchronies: **a** neurotypical (CON, control), **b** dyslexia (DYS), and **c** ASD. Individual asynchronies were plotted with respect to each participant's mean asynchrony, yielding a mean of 0 ms. Consecutive asynchronies are positively correlated in all groups. This positive correlation is largest in the ASD group, reflecting reduced online error correction. Luminance scale is equal in (**a–c**): white, the maximum number of asynchronies in a bin, is 165 in all graphs. **d** Single participant correlations also show the impairment in error correction for the ASD group compared with the neurotypical and dyslexia groups. The median of each group is denoted as a line of the same color; error bars around this median denote an interquartile range. Kruskal–Wallis $H$-statistic and the corresponding $p$ value are plotted in the bottom-left corner; $p$ values of comparisons between groups are plotted next to the line connecting the groups' medians. $N = 109$ subjects ($N_{CON} = 47$, $N_{DYS} = 32$, $N_{ASD} = 30$). Source data are provided as a Source Data file.

indicating that participants partially carry errors across consecutive beats. Calculating single participant correlations (Fig. 2d, median [interquartile range]: neurotypical: 0.52 [0.23], dyslexia: 0.51 [0.27], autism: 0.69 [0.28], Kruskal–Wallis test $H(2) = 8.86$, $p = 0.012$), we found the largest correlation in the autism group, indicating that they retain uncorrected errors longer than the other two groups. The difference between the groups was significant, and post hoc comparisons showed that this is the result of a significant difference between the ASD group and both the neurotypical ($p = 0.033$, Cliff's delta = 0.35) and the dyslexia groups ($p = 0.017$, Cliff's delta = 0.39). The source of reduced error correction between consecutive trials in ASD could be slow perceptual updating, leading to a smaller perceived error, or slow updating of motor plans. Our analysis cannot dissociate between these alternatives.

To understand the dynamics of phase correction we used an autoregressive model to predict the current asynchrony. We consider linear dependencies not only with the previous asynchrony but with several previous asynchronies. We used stepwise regression to determine the number of previous asynchronies to use in the model. We ran the models both at the group level (using separate regressors for each participant but using a group level criterion when adding predictors) and at the single-participant level. The final model included three predictors for all three groups and one to three predictors for 103/109 participants (Supplementary Fig. 4a). That is, it was sufficient to use asynchronies up to three taps back to predict the current asynchrony, and no additional information was given by adding more asynchronies as predictors. There was no difference

between the groups with regard to the number of predictors in the final model ($\chi^2(2, N = 109) = 8.22$, $p > 0.4$). Together, this suggests that phase correction relies only on the most recent information (<2 s). In accordance with the results of Fig. 2, we found a significant difference between the groups in the contribution of the most recent asynchrony to the current asynchrony (Kruskal–Wallis test $H(2) = 6.16$, $p = 0.046$; Supplementary Fig. 4b), indicating that the ASD groups corrected less of the most recent error and carried a larger fraction to the next tap (for more details see Supplementary Note 2 and Supplementary Fig. 4).

*Modeling isochronous tapping reveals that rate of error (phase) correction is slow in ASD.* Impaired phase correction does not rule out that individuals with autism also have noisier representations of the metronome tempo (timekeeper noise), or "sloppier" production of motor commands (motor noise). To address this possibility, we used a well-established computational model of sensorimotor synchronization[28,31,32]. This model assumes that each tapping interval is the summation of three components: timekeeping of external tempo[29,38], the time required for motor execution (both incorporating Gaussian noise), and fraction of perceived error (asynchrony) correction from the previous tap (relative to the mean asynchrony which participants view as synchronous with the metronome). Formally, the model can be written as follows (see Fig. 3a):

$$r_k = -\alpha e_{k-1} + T_k + M_k - M_{k-1} \tag{1}$$

where $r_k$ is the inter-tap-interval of the participant, between

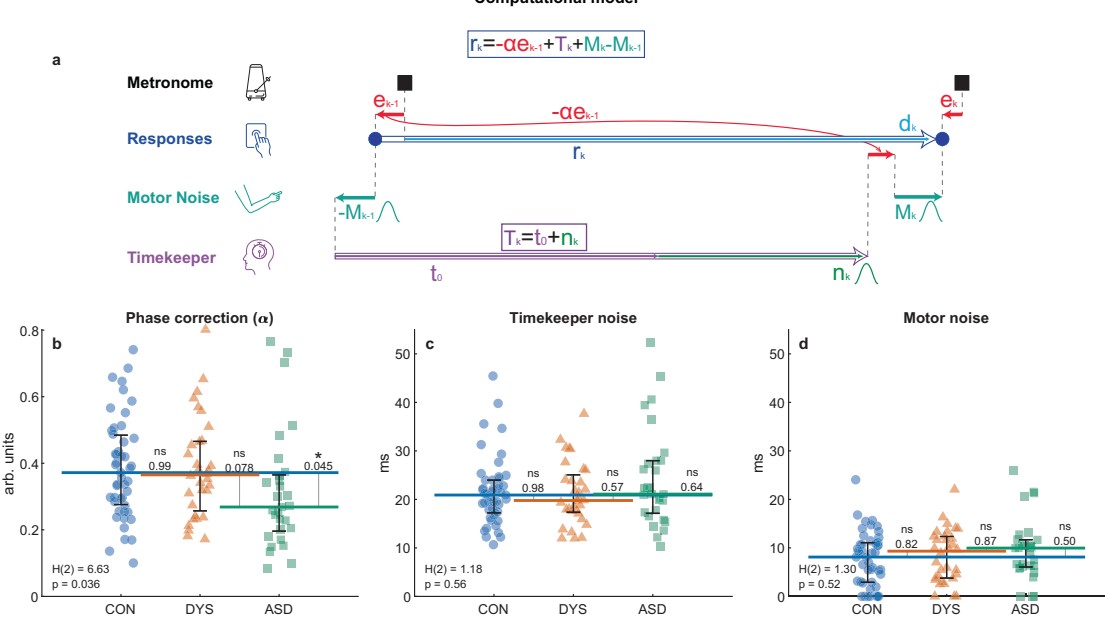

**Fig. 3 Trial-by-trial computational modeling of isochronous tapping: Parameters estimated for each participant show that individuals with autism have reduced error correction and intact timekeeper and motor noise. a** Schematic illustration of the computational model used to dissociate error correction mechanisms from poor timekeeping or motor noise[29,31,32]. Each tapping interval (blue empty arrow) is assumed to be the summation of three mechanisms: (1) error correction based on the previous asynchrony (marked in red, the magnitude of the correction is determined by the phase correction parameter α) (2) timekeeping of the base tempo $T_k$ (composed of a fixed $t_0$, purple, plus the noise at tap k, $n_k$, green), and (3) motor noise (turquoise). See also notations in Fig. 1a. Fitting was performed using the bGLS (bounded General Least Squares) estimation method[28]. **b** Error correction of phase difference—the fraction corrected (α) is significantly smaller in the ASD group. **c** Noise in keeping the metronome period, and **d** Motor noise do not differ between the groups. **b**–**d** Each block was modeled separately, and parameters were averaged over the two assessment blocks. The median of each group is denoted as a line of the same color; error bars around this median denote an interquartile range. Kruskal–Wallis H-statistic and corresponding p value are in the bottom-left corner; p values of comparisons between groups are next to the line connecting the groups' medians. CON control (neurotypical), DYS dyslexia, ASD autism. N = 108 subjects ($N_{CON} = 47$, $N_{DYS} = 32$, $N_{ASD} = 29$), one ASD participant was excluded due to a large number of missing taps (see Methods). Source data are provided as a Source Data file.

metronome beats $k$ and $k-1$, $T_k$ is the participant's current representation of the metronome tempo, $M_k$ is the time of the motor response at time $k$ (both including noise, which is referred to as timekeeper noise and motor noise, respectively), $e_{k-1}$ is the asynchrony at beat $k-1$ and $\alpha$ denotes the proportion of correction of this asynchrony in tap $k$. To maintain a constant asynchrony, positive asynchrony deviations should be followed by shorter intervals and vice versa. Therefore, correction of the next interval is performed by subtracting the magnitude of the current deviation from the estimated tempo, which is why $\alpha$, the phase correction parameter, appears with a negative sign. When $\alpha = 0$ there is no correction and the previous asynchrony is carried to the next response, therefore, larger phase correction will correspond to improved performance on the task.

Note that we can separate the timekeeper component $T_k$ into a fixed mean ($t_0$), which is assumed to be equal to the external metronome tempo, and a noise component with variance $\sigma_T^2$ and zero mean (denoted by $n_k$), such that: $T_k = t_0 + n_k$ (see Fig. 3a). Previous work suggested that the motor noise, associated with each movement onset, and the timekeeper noise, associated with inter-beat intervals, can be distinguished from one another based on the covariance structure of the noise term[29,31,32] (see Methods). Parameter recovery analysis showed a high correlation between the fitted values and the parameters used to generate simulated data (Spearman correlations were larger than 0.92 for all parameters in each of the three groups), indicating that the fitting procedure was highly reliable (Supplementary Note 3 and Supplementary Fig. 6).

We fitted the model for each participant separately and compared the group parameters (Fig. 3b–d). Phase correction was (median [interquartile range]) 0.37 [0.21] in both the neurotypical and dyslexia groups, indicating that error was only partially corrected across consecutive taps, in line with the positive correlation we found (Fig. 2). Yet, phase correction was even smaller (0.27 [0.17]) in the autism group, with a significant group difference (Fig. 3b; Kruskal–Wallis test $H(2) = 6.63$, $p = 0.036$). Post hoc analysis showed a significant difference between the neurotypical and autism groups ($p = 0.045$, Cliff's delta = 0.31) and a marginal difference between dyslexia and autism groups ($p = 0.078$, Cliff's delta = 0.32), but no difference between the neurotypical and dyslexia groups ($p > 0.95$). In contrast to phase correction, we found no group difference in the levels of timekeeping and motor noise (Fig. 3c, d; timekeeper noise (median [interquartile range] (ms)): neurotypical: 20.9 [6.8], dyslexia: 19.8 [7.7], autism: 21.1 [10.8]; Kruskal–Wallis test $H(2) = 1.18$, $p = 0.56$; motor noise (median [interquartile range] (ms)): neurotypical: 8.1 [8.2], dyslexia: 9.3 [8.6], autism: 10 [5.6]; Kruskal–Wallis test $H(2) = 1.3$, $p = 0.52$). The specificity of the group difference to phase correction shows that the larger variability in the autism group does not stem from an elevated noise level in either motor or tempo keeping processes. Importantly, simulations based on the model fitted values per participants reproduced the pattern of differences observed for consecutive correlation values (Supplementary Note 4 and Supplementary Fig. 7).

## Experiment 2—tempo switches reveal reduced online updating of external changes in ASD.

In the second finger-tapping experiment we asked whether individuals with autism or individuals with dyslexia have difficulties in adapting to changing environments. We tested this by switching the tempo of the auditory metronome, so that within each block the tempo alternated between two options (randomly every 8–12 intervals). We quantified the dynamics of updating to the new tempos in our three groups using both model-free and model-based analyses.

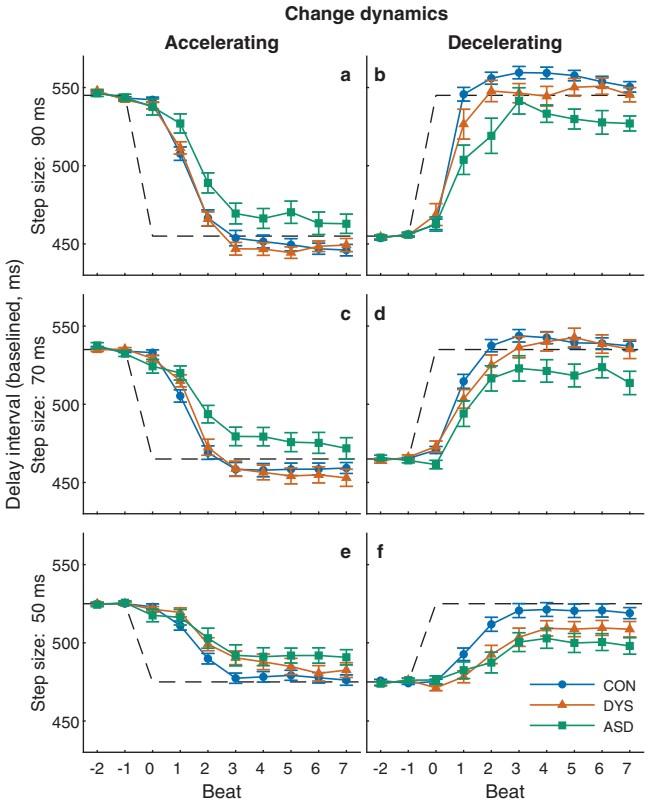

**Change dynamics**

**Fig. 4 Individuals with autism adapt to changes in tempo only partially, even when changes are very salient. a**, **b** 90 ms step-size, **c**, **d** 70 ms step-size, and **e**, **f** 50 ms step-size. In each panel, the x-axis represents the metronome-beat number around the moment of tempo change (beat 0), and the y-axis measures the delay interval in each beat aligned to the pre-change metronome (mean group values, ±SEM; values were calculated by first averaging responses within each participant and then across the group; error bars denote SEM across participants). The dashed lines represent the metronome beat. Changes are quickly corrected, particularly for the larger steps (panels **a**–**d**). Reduced updates are seen for the smaller 50 ms step changes (panels **e**, **f**), where neurotypicals (CON control) take three–four steps to correct, and individuals with dyslexia (DYS) take longer, perhaps since these steps are less salient. The difficulties of individuals with autism (ASD) are seen in all step changes (including the smallest step-size, panels **e**, **f**), and their error is not fully corrected even within seven taps. Each participant tapped through eight-ten accelerations and eight-ten decelerations in each condition. Sample sizes: **a**, **b** 90 ms step-size: $N_{CON} = 46$, $N_{DYS} = 31$, $N_{ASD} = 29$ for acceleration and $N_{ASD} = 26$ for deceleration. **c**, **d** 70 ms step-size: $N_{CON} = 47$, $N_{DYS} = 31$, $N_{ASD} = 29$ for acceleration and $N_{ASD} = 27$ for deceleration. **e**, **f** 50 ms step-size: $N_{CON} = 47$, $N_{DYS} = 32$, $N_{ASD} = 29$ for acceleration and $N_{ASD} = 25$ for deceleration. See Methods for the exclusion criteria. Source data are provided as a Source Data file.

*Individuals with ASD fail to adapt to fast changes in the environment.* Figure 4 shows the timing of tapping in each population aligned to the onset of tempo change (left–acceleration, right–deceleration). We present performance using the delay interval $d_k$ (the time interval from the previous metronome stimulus to the following finger tap, illustrated in Fig. 1a), rather than inter-response interval ($r_k$), since the delay interval uses a constant reference point (the previous metronome beat), whereas the inter-response-interval depends on the previous asynchrony which varies from tap to tap. For presentation purposes, we aligned the pre-change delay interval with the metronome beat (canceling the difference that originated from negative mean

asynchrony, which varies across individuals). The delay interval in the first beat after the tempo change (beat 0) resembles that of the pre-change delay, since the tempo change at this point was not predicted. Following this initial surprise, participants updated their delay intervals to align with the new metronome tempo. This update was faster in the larger and more salient tempo changes[39,40]: in the 90 ms step-size (Fig. 4a, b), which is very salient, the neurotypical and dyslexia groups managed to synchronize to the new tempo after 1–2 metronome beats. This was not the case for the ASD group, which under-corrected in the first and second taps following the change and did not fully adapt even after seven taps. Though this effect is clearest for the 90 ms step-size, similar dynamics can be seen also in the 70 ms step-size (Fig. 4c, d). The smaller, 50 ms step-change (Fig. 4e, f), was less salient and took marginally longer to adapt also for the dyslexia group compared with the neurotypical group, though the difference was not significant in any of our analyses (see following sections). The sluggish update in dyslexia is manifested only in the small tempo change, suggesting that large and abrupt changes are not more challenging to individuals with dyslexia, who do not show an updating difficulty, but possibly reduced perceptual sensitivity to small interval changes. The interpretation of reduced sensitivity to temporal durations, perhaps due to reduced benefits from repeated intervals, is in line with previous observations[41].

*Individuals with ASD do not fully update to tempo changes even following several seconds.* To assess whether updating was attained several beats after the tempo change, we calculated the distributions of the delay intervals in each of the metronome tempos, excluding the four beats immediately after the tempo change, where most tempo update takes place, as shown in Fig. 4 (taking out two–six beats after the change produced similar results). If participants eventually adapt to the change in tempo, the two distributions should be highly separable. This was quantified using measurements from signal detection theory: sensitivity index (d′) and area under the curve (AUC) of the receiver operating characteristic (ROC). In the 90 and 70 ms step-sizes (Fig. 5a–h) we received comparable measurements for the neurotypical and dyslexia groups, and reduced values for the autism group, though in the 50 ms step-size (Fig. 5i–l) the values of the dyslexia group were between those of the neurotypical and autism groups. This pattern was replicated when we looked at single participant values: on both measures (d′ and AUC), there was a significant difference between the groups in all conditions (Kruskal–Wallis test; all $p < 0.012$). Post hoc comparisons showed a significant difference between the autism and neurotypical groups in all step-sizes and measures (all $p < 0.008$), and between the autism and dyslexia groups in the larger step-sizes. The difference between the neurotypical and dyslexia groups was not significant in any step-size (all $p > 0.4$).

Importantly, d′ and AUC are both affected by the SD of the distribution of asynchronies. Since the SD in the ASD group is larger (Experiment 1), normalizing by SD would decrease d′ in this group more than in the other groups. In order to see if there is an impairment in the autism group on top of the increased variability, we used the difference between the means of the distributions without SD normalization. We found comparable values for the neurotypical and dyslexia groups, and smaller values in the ASD group, for the 90 and 70 ms step-size conditions (Fig. 5a–h). Looking at single participants this pattern was preserved (Kruskal–Wallis test 90 ms: $p = 0.007$; 70 ms: $p = 0.014$), with post hoc comparisons showing significant differences between the neurotypical and autism groups, and between dyslexia and autism groups (all $p < 0.05$), with no

difference between neurotypical and dyslexia ($p > 0.6$). For the 50 ms step-size (Fig. 5i–l) we found that the dyslexia group value is midway between that of the neurotypical and the ASD groups, as in other measures of small tempo changes (single participant Kruskal–Wallis test: $p > 0.2$). Combined measures (formed by z-scoring each step-size condition using the mean and SD of the neurotypical group, then averaging over the different conditions) showed a significant difference between the groups in all measures (Kruskal–Wallis test, $p < 0.002$ for d′ and AUC and $p = 0.013$ for the difference of means), and post hoc comparisons showed no differences between the neurotypical and dyslexia groups (all $p > 0.4$), but significant differences between the neurotypical and ASD groups ($p < 0.001$, Cliff's delta $> 0.45$ for d′ and AUC and $p = 0.017$, Cliff's delta $= 0.37$ for the difference of means) and between dyslexia and ASD groups ($p = 0.04$ for AUC and difference of means, $p = 0.08$ for d′, all Cliff's delta $> 0.35$).

*Modeling the parameters underlying tempo switches reveals slow period-updating in ASD.* In Experiment 1 the mean timekeeper period was assumed to be a fixed value ($t_0$)—the metronome period. To model changing environments, we now enabled changes in the mean estimate of timekeeper, so that instead of decomposing $T_k$ into a fixed mean and a noise component as we did in the isochronous case, we use the following equation:

$$T_k = t_k + n_k \qquad (2)$$

where $t_k$ is dynamically adapting to the changes in tempo. The estimate of the tempo should be informed by the asynchrony, where large positive errors indicate an acceleration in tempo (the period getting shorter), so the internal estimate must be reduced, and large negative errors indicate a deceleration. We used a model proposed by Schulze et al.[42], where this intuition regarding tempo correction is implemented using the following equation (Fig. 6a):

$$t_k = t_{k-1} - \beta e_{k-1} \qquad (3)$$

Where $\beta$ is a parameter denoting the proportion of correction of the period estimate for interval $k$. Optimally, the period estimate should track the changes in the external tempo, but it would not be an ideal strategy to change this internal estimate too rapidly, since asynchrony errors can result from noise in the participant's taps. The magnitude of $\beta$ determines the pace of this updating procedure. The full model is defined by the equation of the model of Experiment 1 (Eq. (1), Fig. 3a), substituting Eq. (2) coupled with the equation for the dynamics of the period correction (Eq. (3), Fig. 6a):

$$r_k = -\alpha e_{k-1} + t_k + n_k + M_k - M_{k-1} \qquad (4)$$

$$t_k = t_{k-1} - \beta e_{k-1}$$

To disentangle the estimates of the phase correction ($\alpha$) and period correction ($\beta$) we use the bGLS method[28] (see Methods). To enhance the model's sensitivity to the changes, we used only the segments immediately before and after the tempo change. We fit the model to each tempo-change segment separately and averaged the resulting parameter values for each step-size (first per block, and then across blocks). The extended model explained the data of Experiment 2 substantially better than a model without period correction, namely the model of Experiment 1 (likelihood ratio test, $p < 0.001$ for all subjects and Akaike information criterion (AIC) for the extended model is smaller than the original model for all subjects). Adequate parameter

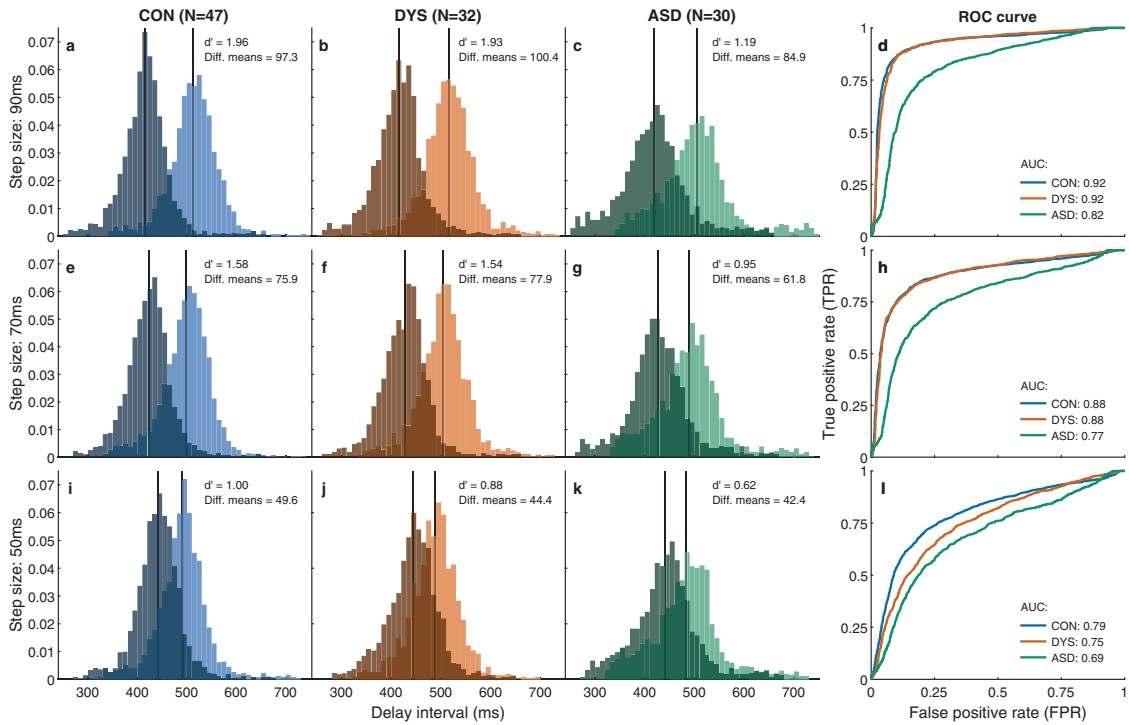

**Fig. 5 Distributions of delay intervals 5–12 taps after the tempo switch show that asynchronies in the autism group remain uncorrected. a–c**, **e–g** and **i–k** show group delay intervals probability density functions separately for the longer tempo (light color) and shorter tempo (dark color), for each tempo-change condition (90 ms—top, 70 ms—middle, 50 ms—bottom) and for each population (CON (control, neurotypical)—blue, DYS (dyslexia)—red, ASD (autism)—green). The mean of each distribution is denoted by a black vertical line. Values of d′ and the difference between the means (diff. means) are in the top right corner. **d**, **h**, **l** show for each group the receiver-operator-curves (ROC), and area under the curve (AUC) for classification between delay intervals under the two tempos. In the 70 and 90 ms step-sizes, the measures of dyslexia and neurotypical groups nearly overlap, while the values for the ASD group are smaller, reflecting reduced updating to changes in external tempo. For the small 50 ms step-size, the dyslexia group values are midway between the neurotypical and autism values, though this difference was not significant (see main text). Source data are provided as a Source Data file.

recovery is shown in Supplementary Fig. 6 and Supplementary Note 3.

In each of the step-size conditions, we found a significant group difference in period correction (Kruskal–Wallis test, all $H(2) > 8$, $p < 0.018$), with no significant differences in the other parameter estimates (Kruskal–Wallis test, all $H(2) < 3.3$, $p > 0.2$). Since the optimal values for error correction depend on context, we obtained combined estimates by z-scoring each parameter for each step-size condition (using the mean and SD of the neurotypical group) and averaging over the different conditions (Fig. 6b–e). As expected from the single condition results, the ASD group had a significantly smaller period correction (z-scored $\beta$, median [interquartile range]: neurotypical: 0.13 [0.95], dyslexia: −0.29 [1.46], autism: −1.03 [1.32]; Kruskal–Wallis test $H(2) = 15.59$, $p = 0.0004$, Fig. 6b). Post hoc comparisons showed a significant difference between the autism and neurotypical groups ($p = 0.0002$, Cliff's delta = 0.54), and between the autism and dyslexia groups ($p = 0.048$, Cliff's delta = 0.35), with no difference between the neurotypical and dyslexia groups ($p = 0.34$). No differences were found in other estimated parameters (all $p > 0.16$, see Fig. 6c–e), including z-scored phase correction ($\alpha$). Simulations based on the fitted values of each participant were able to reproduce the observed patterns of reaction to changes that were characteristic of each group (compare Fig. 4 to Supplementary Fig. 8, Supplementary Note 4).

To conclude, individuals with autism show reduced initial updating of tempo, which is not fully corrected within the next 3–4 s (>7 taps), as can be seen in Figs. 4, 5.

Having found group differences in phase correction in a stationary environment ($\alpha$, Experiment 1) and in period

correction in the changing-tempo protocol ($\beta$, Experiment 2) we asked whether these two parameters denote separate mechanisms, or, alternatively, both reflect the same mechanism of online error correction. In a tempo-change paradigm, the relative contributions of the processes of correction for phase error and for period error are difficult to dissociate, since these errors are temporally correlated[39,43]. The large errors immediately following the tempo change are always the summation of the error directly induced by the metronome's tempo change (which requires a genuine period correction), and the error induced by the participant's inability to predict the point of tempo change (inducing an additional step-change phase error at beat zero). To resolve this ambiguity, we assessed the cross-participant correlation between the parameter of phase correction in Experiment 1 (Fig. 3b), and period correction in Experiment 2 (Fig. 6c). We found significant positive correlations in each of the three groups separately (Spearman correlations: $\rho_{CON} = 0.44$ ($p < 0.002$), $\rho_{DYS} = 0.5$ ($p < 0.005$) and $\rho_{ASD} = 0.61$ ($p < 0.001$), Fig. 7a–c) and when combining the groups ($\rho_{ALL} = 0.55$ ($p < 0.001$)). By contrast, there were no significant correlations between the other error correction parameters in any of the three groups (all $|\rho| < 0.18$, $p > 0.35$, for the correlations between the two error terms of Experiment 2, and the two estimations of phase correction). This combined pattern of correlations suggests that phase correction in Experiment 1 and period correction in Experiment 2 are manifestations of a common mechanism of online error correction. We, therefore, formed a combined update rate score by averaging the correction parameters of both experiments (again after z-scoring with respect to the neurotypical group). Update rate showed a significant difference between

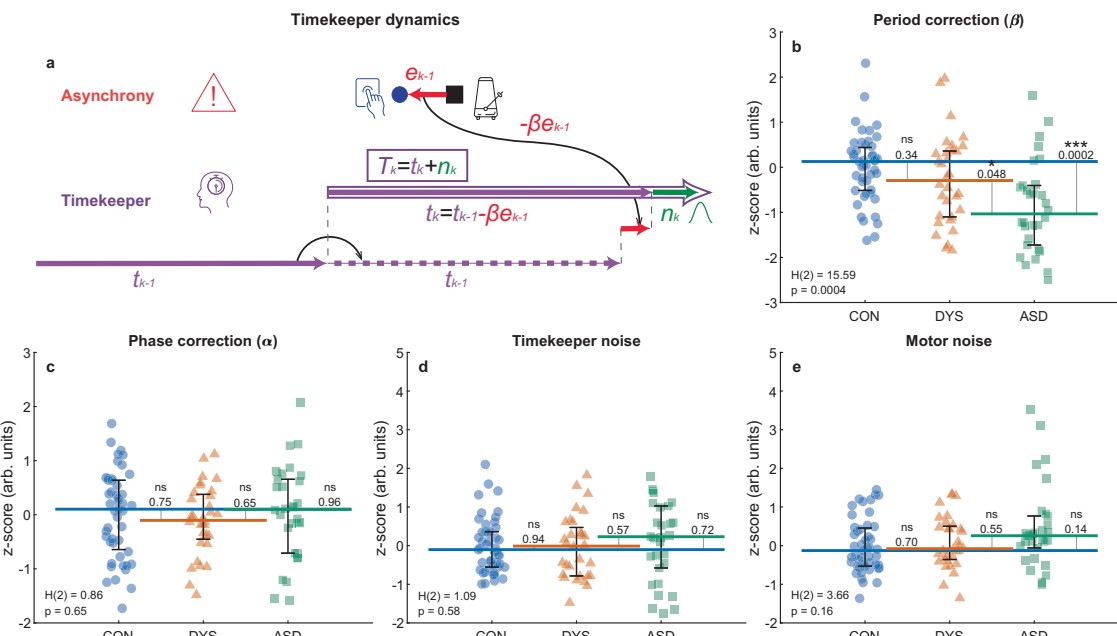

**Fig. 6 Trial-by-trial computational modeling of tapping with changing tempos: Parameters estimated for each participant show that individuals with ASD have reduced period correction when the period changes abruptly. a** Illustration of the dynamics added to the isochronous computational model (Fig. 3a), which enables tracking of changing tempos[42]. The internal period estimate ($t_k$, purple) is adjusted in each trial based on the recent asynchrony (in red, the magnitude of the correction is determined by the period correction parameter $\beta$). To produce the tapping interval, noise is added to this estimate ($n_k$, green) as in the isochronous model. **b–e**: Each dot represents the combined value from all conditions of tempo-switches per participant (50–90 ms; after z-scoring). CON control (neurotypical), DYS dyslexia, ASD autism. $N = 109$ subjects ($N_{CON} = 47$, $N_{DYS} = 32$, $N_{ASD} = 30$). **b** Period correction ($\beta$) is smaller in the autism group compared with the two other groups, **c** while phase correction ($\alpha$) is similar in this Experiment. **d** Timekeeper noise and **e** Motor noise estimates do not differ between the groups. The median of each group is denoted as a line of the same color; error bars around this median denote an interquartile range. Kruskal–Wallis $H$-statistic and corresponding $p$ value are in the bottom-left corner; $p$ values of comparisons between groups are next to the line connecting the groups' medians. Source data are provided as a Source Data file.

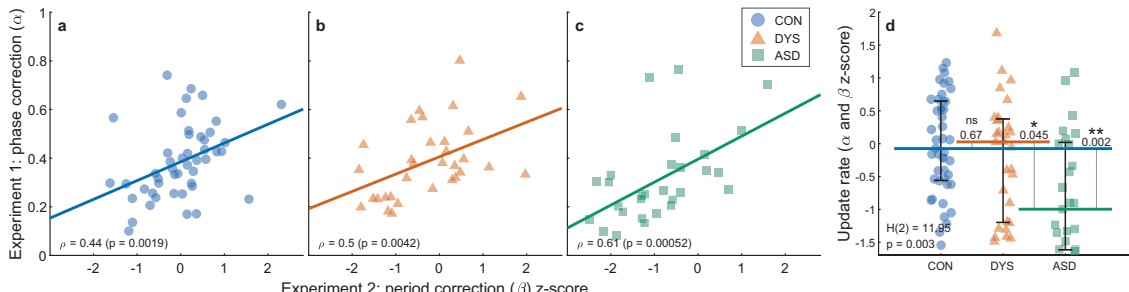

**Fig. 7 Rate of online error correction in stationary and in changing environments reflect a single updating mechanism.** The estimated phase correction from Experiment 1 and the estimated period correction from Experiment 2 are highly correlated in all groups **a** neurotypical (CON control), **b** dyslexia (DYS), **c** autism (ASD), suggesting that both are manifestations of a common underlying mechanism of error correction, which determines the speed of integrating new sensory data to guide behavior. The significance of Spearman correlation was calculated using a two-sided test, $p$ values are uncorrected. Overlayed regression lines, predicting phase correction (Experiment 1) from period correction (Experiment 2) with an intercept term. **d** The combined update rate is significantly smaller in the ASD group but does not differ between the neurotypical and dyslexia groups. The median of each group is denoted as a line of the same color; error bars around this median denote an interquartile range. Kruskal–Wallis $H$-statistic and the corresponding $p$ value are plotted in the bottom-left corner; $p$ values of comparisons between groups are plotted next to the line connecting the groups' medians. $N = 108$ subjects ($N_{CON} = 47$, $N_{DYS} = 32$, $N_{ASD} = 29$), one ASD participant was excluded from the computational modeling of Experiment 1 due to a large number of missing taps (see Methods). Source data are provided as a Source Data file.

the groups (median [interquartile range]: neurotypical: $-0.07$ [1.21], dyslexia: 0.03 [1.57], autism: $-1$ [1.63]; Kruskal–Wallis test $H(2) = 11.95$, $p < 0.003$, Fig. 7d). Post hoc comparisons revealed a significant difference between the neurotypical and ASD groups ($p < 0.002$, Cliff's delta = 0.45), and between dyslexia and ASD groups ($p = 0.045$, Cliff's delta = 0.35), with no difference between the neurotypical and dyslexia groups ($p > 0.65$, Cliff's delta = 0.12). Overall, the autism group had a

substantially lower updating rate yielding slower correction rates in both fixed and changing environments.

*Update rate is correlated with communication and mindreading skills.* Since previous literature suggests that synchronization is associated with social skills[17,18], we asked whether slower updating is correlated with these skills among our participants in the neurotypical and autism groups. We administered to

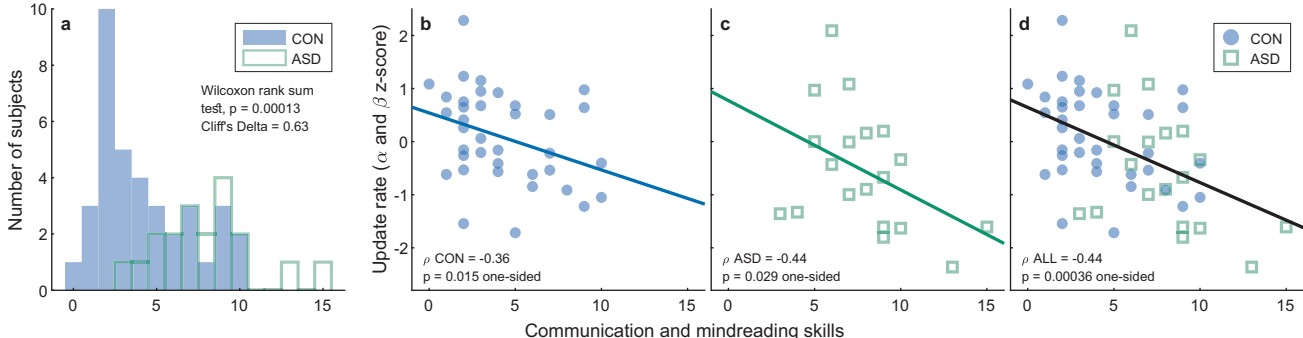

**Fig. 8 Update rate is correlated with communication and mindreading skills in both autism and neurotypical groups. a** Distribution of communication \mindreading skills was measured using a factor of the AQ50[44,45] in the neurotypical (CON control) and autism (ASD) groups. **b–d** Update rate is correlated with communication\mindreading skills in the neurotypical (**b**) and autism (**c**) groups, and when combining both groups together (**d**). Each dot represents a single participant (neurotypical $N_{CON} = 37$, ASD $N_{ASD} = 19$). Lines represent regression lines predicting update rate from communication and mindreading skills with an intercept term. Blue and green lines (**b**, **c**) are based on the neurotypical and autism groups respectively, the black line (**d**) is based on data from both groups combined. Source data are provided as a Source Data file (**b–d**).

participants in both groups the AQ50 (Autism Quotient)—a self-report questionnaire, aimed to assess the severity of autism-related traits[44]. Nineteen of our participants with autism and 37 neurotypical participants filled the questionnaire. The questionnaire is composed of several subscales which together assess several traits associated with autism, including social and communication skills. Higher scores on the questionnaire indicate more autistic traits. Accordingly, we found significantly higher scores in the autism group (median [interquartile range]: 75 [23.5]) compared with the neurotypical group (median [interquartile range]: 52 [17.5]), Wilcoxon rank-sum test, $p < 0.001$, Cliff's delta $= 0.69$.

We hypothesized that a slower update rate (the combined $z$-score of $\alpha$ in Experiment 1 and $\beta$ in Experiment 2, Fig. 7d) would correspond to poorer social and/or communication skills. We used the three-factor model of the AQ50 proposed by Austin[45], which separates between individuals' cognitive social abilities (theory of mind—their ability to understand other people's thoughts, communication/mindreading factor), and their emotional propensities (joy from being with others and socializing, social skills factor). The combined update rate was not correlated with the social skills factor in either group ($\rho < 0.12$ for neurotypicals and $p > 0.5$ for individuals with ASD). However, it was significantly correlated with the communication factor in both the neurotypical (Fig. 8b, Spearman correlation $\rho_{CON} = -0.36$; $p < 0.03$) and the ASD groups (Fig. 8c, Spearman correlation $\rho_{ASD} = -0.44$, in a two-tailed test $p = 0.058$, in a one-tailed $p < 0.029$, which is justified based on our a priori hypothesis). Importantly, despite the large group difference in the communication factor (Wilcoxon rank-sum test, $p < 0.0002$, Cliff's delta $= 0.63$, Fig. 8a), the neurotypical and ASD groups showed a similar pattern of correlation between communication skills and updating rate (Fig. 8b, c). Bootstrap permutations showed that the correlation values of the two groups were not significantly different ($p = 0.78$), and both could be approximated using the correlation in the combined group (see Methods, section AQ50 questionnaire). We therefore also assessed the correlation across both groups (Fig. 8d), which was highly significant (Spearman correlation $\rho_{ALL} = -0.44$, two-tailed test $p < 0.0015$ (Bonferroni corrected for two factors)).

## Discussion
We found that individuals with autism fail to synchronize their movements to external cues, unlike individuals with dyslexia, who are able to synchronize adequately. Using trial-by-trial computational modeling, we were able to precisely pinpoint the underlying deficit: we found that the level of noise in both motor processing and internal timekeeping is sufficient in individuals with autism, yet they use recent sensory information to a lesser degree when compared with the other two groups. Consequently, they are slower to correct their synchronization errors (Experiment 1) and are slower to adapt their internal representation to changes in the environment (Experiment 2).

To understand the pattern of deficits found in the autism group we used a well-established model of sensorimotor synchronization[31,32]. In this model, each tap is informed by two distinct sources of prior information: a long-term source, the timekeeper, holding information about the distribution of inter-beat-intervals accumulated over the experiment; and a short-term source, responsible for online error correction, that relies on the most recent asynchronies. Together, the mean value of the timekeeper (the metronome tempo) and the error of the most recent tap provide prior information for performance in the current trial. The long-term component (the mean value of the timekeeper) is reliably kept by the participants with autism, whereas the recent information, which needs to be quickly integrated into the timing of the next tap either due to inherent noise in motor execution or to a sudden tempo change, is used less in individuals with autism than in the neurotypical and dyslexia groups, suggesting a slower integration rate.

This observation suggests an underweighting of recent sensory information into a form that can be used to guide behavior, in line with the "slow updating" framework[11]. Importantly, in this study participants had a strong incentive to utilize recent sensory information, which always improved synchronization, but individuals with autism nonetheless failed to do so. The "slow updating" framework proposes that Bayesian integration will be impaired in autism when fast integration is needed but will otherwise be intact. This stands in contrast to the predictions of "increased volatility" accounts, which propose that individuals with autism overestimate the volatility of the environment[10,12], or that individuals with autism overweigh their prediction errors[9]. According to these accounts, individuals with autism evaluate the environment's statistics as changing more frequently than it actually does, and therefore they would be expected to quickly update their internal model to meet their estimated rate of environmental change. We directly tested this prediction by using blocks with changing tempos and found reduced updating in the autism group, rather than accelerated updating, in line with the "slow updating" framework.

The slow-update conceptualization explains many seeming inconsistencies in the literature assessing motor performance,

sensorimotor performance, and even finger tapping in ASD. This literature characterized motor skills but did not study the rate of online updating as the limiting bottleneck. For example, when individuals were asked to keep tapping after the metronome stops (unpaced tapping), the performance of the ASD group was comparable to neurotypicals'[46,47]. This seems surprising, since these conditions are more cognitively demanding[48]. However, the constraint on performance here is keeping the previous tempo, i.e., the robustness of working memory rather than synchronization with external stimuli. In such conditions, slow-update counterintuitively predicts that the performance of individuals with autism will be similar to that of neurotypical controls, since serial online error correction is not a limiting bottleneck. This is indeed the observation[46,47] and is also consistent with our finding of similar timekeeper noise in the three populations (Figs. 3c, 6d). Similarly, in demanding tasks that require more complicated learning mechanisms, and hence do not rely on online error correction, individuals with ASD are expected to show typical performance, which is indeed the case[49]. However, when test conditions require online synchronization, their performance manifests elevated variability[15]. Interestingly, in line with our findings of reduced serial error correction in the ASD group, error-related negativity (ERN) event potential has a lower amplitude and longer latency in ASD[50,51]. This ERP component is also associated with the correction of large asynchronies in finger tapping[52].

Our analyses also suggest a mechanistic account to the motor "clumsiness", reported already in early descriptions of autism[53], and commonly observed since[54,55]. We find that motor function is not inherently noisy in autism, but rather, that the process of integrating sensory information into motor plans is slower. Hence, while there is an essential sensory component to many movement forms, we expect individuals with autism to experience the greatest difficulty when fast integration is required. This prediction is supported by recent reviews analyzing the core difficulties underlying poor sensorimotor integration in autism[4,5]. Whyatt & Craig[4] show that the motor deficit in autism is specific to tasks requiring fast sensorimotor integration, for example, individuals with autism show a deficit in catching a ball, which requires rapid integration of visual information, while they show intact throwing, which is internally driven. Both reviews suggest that impaired sensorimotor integration may underlie all deficits found in autism spectrum disorder. We propose that impaired sensorimotor integration stems from reduced use of sensory evidence to correct for errors, which is a specific manifestation of slow updating of internal models[11].

The specific stage of processing which yields the slow update is difficult to pinpoint. The slower processing stage could transpire at the perceptual level, in which case the motor manifestations are inherited. Namely, the tapping task relies on fast and accurate error calculation, which require fast comparisons between the timing of the external metronome and the proprioception of the finger tap. If cross-modal integration is sloppier in autism, or temporal windows are less precise[3,56,57], then perhaps occasionally no error is calculated, leading to a bias of underestimating the error, and consequently to reduced synchronization. In our model, it would lead to smaller alphas and betas. A recent study using Bayesian modeling to understand the deficits of individuals with autism in a visual path integration task can also be understood within this framework. Noel et al.[58] found significantly larger variability in motor execution in the autism group, and their modeling framework revealed that individuals with autism are impaired in scaling their sensory likelihood function when executing the next action. Inadequate scaling can be a sign of poor updating of priors but can also stem from impairments at the sensory level.

We should note however that in both Noel et al.'s path integration and our paced finger tapping task, the impaired use of sensory information was measured in conditions of serial actions, where adequate performance requires fast integration of sensory information to inform the next behavior. In conditions where trials are embedded in a setting that does not rely on fast cross-trial or cross response updates, the responses of individuals with autism are typically fast and temporally accurate[59,60]. For example, assessing temporal estimation, Edey et al.[61] presented participants with four auditory (or visual) stimuli with equal temporal intervals in each trial and asked participants to listen to the first two stimuli and press a button in temporal alignment with the third and fourth. The temporal accuracy of participants with autism was similar to that of neurotypical participants and even better than neurotypicals' in the visual task. Adequate perception of tempo is in line with our findings of adequate timekeeper noise. But importantly, their study did not assess serial dependence effects across trials. When serial effects were measured in a task of temporal reproduction, and the impact of previous trials' intervals was assessed, it was found that children with autism underuse previous intervals[62], in line with the "slow updating" framework. A difference in serial dependency profiles between the groups may also underlie the higher accuracy of the autism group in the visual condition observed by Edey et al. It has been shown in several contexts that visual sensorimotor synchronization is noisier than auditory sensorimotor synchronization[23,48,63], which may lead participants, particularly neurotypicals to increase the magnitude of serial dependency[64], and perhaps consequently hamper their performance[11].

Our observation of synchronization difficulties in a nonsocial context indicates that poor synchronization is not a unique outcome of a lack of social interest[2]. Rather, reduced synchronization may reduce the interest in other people's state of mind, though causality is likely to operate in both directions. We found a correlation between our measure of update rate and mindreading skills, in both neurotypicals and people with ASD, yet we did not find a significant correlation with social joy. There is also other evidence for distinct processes underlying the neurocognitive vs. affective influences on social skills[65]. Therefore, it is possible that the update rate taps onto one mechanism, but not all. Further studies, which include direct clinical measures, are needed to clarify the functional relations.

In contrast to the autism group, the dyslexia group had no difficulties in sensorimotor synchronization. This observation is at odds with the temporal sampling framework of dyslexia[66], which posits that individuals with dyslexia have problems with oscillatory entrainment, specifically in the delta range (1.5–4 Hz). The temporal sampling theory predicts impairment in rhythmic motor performance at the tested range of 2 Hz. However, early studies of individuals with dyslexia found no deficit in simple-paced tapping tasks[67,68]. Follow-up studies[69,70] obtained mixed results in paced finger tapping, and difficulties depended on the exact tempo around 2 Hz. Still, we should note that we did find a subtle deficit in the dyslexia group in adapting to small tempo changes (50 ms), though not in the isochronous condition. The specificity of the very mild deficit in dyslexia to small changes in tempo suggests that it reflects a slightly reduced sensitivity to tempo, perhaps due to reduced benefits from interval repetition[41], but we cannot rule out alternative accounts. Though the difference from the neurotypical group was not significant in any of our analyses, in the small tempo change the dyslexia group's performance also did not significantly differ from that of the ASD group.

To conclude, our study compared two prominent computational accounts of autism—the "increased volatility" account and the "slow updating" account. Our results support the "slow

updating" account, which proposes that slow update of internal representations is a core deficit of autism, contributing to both perceptual and motor difficulties. More broadly, our study demonstrates how computational modeling can be used in order to better understand the dynamics of information processing in perception and action in both typical and atypical populations. This approach can lead to the novel integration of computationally informed methods for clinical applications.

## Methods

**Participants**. Neurotypical participants and participants with dyslexia were recruited through advertisements at the Hebrew University of Jerusalem and colleges near the university. Participants with ASD were recruited through clinics (including author T.E.'s clinic), designated facilities, and support groups. Multiple recruitment sources were used to balance any potential biases that each single source might suffer from. All participants in the dyslexia group had been diagnosed by authorized clinicians as having a specific reading disability and all participants with ASD were diagnosed by authorized clinicians and were consequently entitled to Israeli government support aimed specifically for individuals with ASD. All participants were native Hebrew speakers (either born in Israel or immigrated to Israel before the age of 4 years), with no more than minimal musical education (less than 3 years of self-reported musical education). We added the latter restriction (as in ref. [11]) since performance on sensorimotor tasks may be enhanced by musical background[71–73], and may affect clinical groups to a different extent[74]. We recruited participants to all groups within a predefined time period, which was to be extended if one of the groups contained less than 20 participants. By the end of the recruitment period, all groups were larger than 20 participants. Participants with autism were recruited from multiple sources to ensure the sample is representative. All participants completed a set of cognitive assessments, which evaluated general reasoning skills by the standard Block Design task (WAIS-IV[75]) and reading abilities by pseudoword and paragraph reading (details can be found in ref. [11]). They all performed the same protocol of finger tapping—Experiments 1 and 2. Participants in all groups were randomly sampled.

Data were collected from 133 participants (56 neurotypical, 39 dyslexia, and 38 autism). Of these, $N = 24$ ($N_{ASD} = 8$, $N_{DYS} = 7$, $N_{CON} = 9$) were excluded. Our exclusion policy (determined prior to data collection) was aimed to ensure that the general reasoning skills of all participants are no less than two SD below the general population mean (scaled Block Design scores > 6), age, and general reasoning scores are matched across the three groups and reading skills of the neurotypical and ASD groups are matched. Since the focus of this research was the ASD population, we excluded participants in a way which kept the largest number of participants with ASD. Excluding all participants with a Block Design score < 7 excluded one participant with dyslexia, and six with ASD. Matching Block Design scores, while keeping as many participants with ASD as we could, led us to exclude neurotypical and dyslexia participants with Block Design > 15: 7 neurotypical, and four with dyslexia. Reading-related measures (assessed in the lab) led to excluding one neurotypical participant with exceptionally low pseudoword reading (more than 2 SDs below group mean) and two participants with dyslexia exceptionally high pseudoword reading scores (> 2 SDs above group average). Finally, three participants were excluded due to extreme mean asynchrony values (> 3 SDs above the population mean, based on previous studies)—one neurotypical and two in the autism group. The final group consisted of 109 participants (47 neurotypical, 32 dyslexia, and 30 autism). These groups were matched in age and reasoning skills, measured by the standard Block Design task. Results of these assessments are reported in Supplementary Table 1. Importantly, this exclusion policy only weakened the results reported in the paper (namely, the population without the exclusion show larger effect sizes compared with what we report in the paper) since neurotypical participants with higher Block Design scores tend to be better tappers (lower SD, better error correction) and individuals with ASD with lower Block Design scores tend to be poorer tappers. Piloting began in November 2013. The collection of data from the neurotypical and dyslexia participants took place between March 2014 and September 2015, and between May 2017 and March 2018. ASD data collection took place between December 2015 and March 2018.

All experiments were approved by the Ethics Committee of the Psychology Department of the Hebrew University and the Helsinki Ethics Committee of Sheba Hospital (required for testing individuals with ASD recruited through their adult clinic). All participants provided written informed consent and were financially compensated for their time and travel expenses.

**Finger tapping experimental design**. Participants heard a series of metronome beats and were asked to start tapping in synchrony with the metronome. To help participants synchronize, they were instructed to listen to the metronome first and tap after about three metronome beats[23]. The metronome beats were heard through headphones at a comfortable presentation level. Tapping was performed on a custom-made wooden box, including a microphone which recorded the participant's responses. We used either Focusrite Saffire 6 USB or Focusrite Scarlett 2i2 sound cards, which simultaneously recorded the output from the microphone installed inside the box and a split of the headphone signal using the open-source

software audacity (https://www.audacityteam.org/), so that the playback latency and jitter could be estimated for each recording. Onsets were extracted from the stereo audio signal using a custom Matlab script. The overall latency and jitter obtained in this way, measured separately using calibration hardware, was about 2 ms[76].

The task consisted of 12 blocks, each containing ~100 metronome beats. Rhythmic patterns consisted of identical short percussive sounds ("clicks") lasting 55 ms with an attack time of 5 ms generated from amplitude modulated white noise. Blocks were separated by short pauses of 5 s. Participants had two breaks, after the third and eighth blocks. Prior to the test procedure, all participants completed one block of practice. Researchers were present during the demo block, but usually left the room for the experimental session, except in rare cases when the testing conditions did not enable this. The researchers were not blind to the hypothesis or condition during collection.

Blocks were separated into six conditions and each was repeated twice. The first condition (Experiment 1) had an isochronous tempo of 2 Hz—beats were presented with an inter-onset-interval (IOI) of 500 ms, known to be close to the optimal tempo for synchronization[23,77]. The other five conditions (Experiment 2) were composed of alternating tempos. In each block, the metronome tempo alternated between two options, which differed symmetrically from the baseline tempo of the isochronous condition (500 ms): one tempo was faster than this baseline and the other was slower. Metronome changes occurred randomly every 8 to 12 intervals, thus both changes were repeated several times in each block (the design was similar to ref. [43]). We used five different conditions with deviations ranging from ±5 to ±45 ms, in steps of 10 ms: (1) 495 and 505 ms (±5 ms, step-size of 10 ms), (2) 485 and 515 ms (±15 ms, step-size of 30 ms), (3) 475 and 525 ms (±25 ms, step-size of 50 ms), (4) 465 and 535 ms (±35 ms, step-size of 70 ms), and (5) 455 and 555 ms (±55 ms, step-size of 90 ms). Each block contained two types of changes: acceleration (slow to fast tempo change) and deceleration (fast to slow tempo change). For example, in condition (3) the acceleration was a change from 525 to 475 ms and deceleration was the change from 475 to 525 ms. The 12 task blocks (including Experiment 1 and Experiment 2) were presented in one of four pseudorandomized orders.

As explained above, the tempo changes in Experiment 2 covered a broad range and were chosen based on previous literature, which tested musicians or trained participants[43,78]. Our novice, musically untrained participants had markedly higher tracking thresholds—the two smaller step changes were largely unnoticed by our participants (Supplementary Fig. 5). We, therefore, focused our analyses on the three larger step-sizes shown in Figs. 4–7. Importantly, the computational modeling results remain highly significant also when including the smaller tempo changes (Fig. 6: $\beta$ Kruskal–Wallis test $H(2) = 14.13$, $p = 0.0009$. Post hoc comparisons show a significant difference of autism and neurotypical groups ($p = 0.0005$), and a marginal difference between the autism and dyslexia groups ($p = 0.068$); Fig. 7: Spearman correlations between phase correction and period correction $\rho_{CON} = 0.54$ ($p = 0.00012$), $\rho_{DYS} = 0.5$ ($p = 0.0043$) and $\rho_{ASD} = 0.66$ ($p = 0.00015$), and significant group difference in update rate: $H(2) = 11.05$, $p = 0.004$; Fig. 8: Spearman correlations between update rate and communication and mindreading skills: $\rho_{CON} = -0.31$, $\rho_{ASD} = -0.41$, $\rho_{ALL} = -0.38$, all $p < 0.05$, one-sided). We show the mean and SD of the asynchrony in all experiment blocks (including the two smaller step-sizes) in Supplementary Figs. 2 and 3.

**Finger tapping analyses**. All analyses and statistical procedures were performed using Matlab (version 2019b). To measure synchronization, we used the time interval between the metronome stimulus and participant's responses (asynchrony, we denoted it by $e_k$, see Fig. 1a). Participants usually anticipate the metronome beat resulting in a negative mean asynchrony[21,23]. We denote by $r_k$ and $s_k$ the inter-tap-interval and inter-stimulus-interval between taps $k$-1 and $k$, respectively. We denote by $d_k$. the delay interval between metronome beat $k$-1 and the next participant tap (corresponding to beat $k$). Note that $r_k = d_k - e_{k-1}$ (see Fig. 1a).

A model-free characterization of tapping performance in a given block is given by the mean asynchrony and the SD of asynchronies in that block. In Experiment 2 perturbations of the metronome, tempo occurred at unexpected time points, therefore we computed the mean and SD after removing the contribution of the unexpected perturbation ($s_k - s_{k-1}$). Namely, we compute the mean and SD of $e'_k = e_k + (s_k - s_{k-1})$. Results (Fig. 1b, c and Supplementary Figs. 2, 3) were averaged over the two repetitions of each condition.

We excluded response taps that were outside a window of ±200 ms surrounding metronome beats[23]. Omitted taps are cases where the participant did not tap within a 400 ms window around the metronome beat. Overall, there was a small number of omitted or excluded taps—less than 5% of the taps (across experiments). In Experiment 1 the percentage of omitted or excluded taps was (median [interquartile range] (%)): neurotypical: 0.5 [1.3], dyslexia: 0.5 [1.5], autism: 1.2 [4.9]. The difference between the groups was marginally significant (Kruskal–Wallis test, $H(2) = 5.8$, $p = 0.055$), corresponding to our finding of more variable tapping in the autism group. In Experiment 2 the percentages were (median [interquartile range] (%)): neurotypical: 1.3 [2.5], dyslexia: 1.5 [3], autism: 2.8 [10], which is again marginally significant (Kruskal–Wallis test, $H(2) = 5.58$, $p = 0.06$). Computational modeling was performed only on blocks with less than 40% omitted or excluded taps. This excluded three blocks from Experiment 1, and

eight blocks from Experiment 2 (one block from 50 ms step-size, four blocks from the 70 ms step-size, and three blocks from the 90 ms step-size).

*Experiment 1*

## Autocorrelation analysis

As a first approach to assess the rate of error correction we computed the (Pearson) correlations between consecutive asynchronies ($e_k$). For this analysis we used perceived asynchronies, meaning the interval between the current asynchrony and the mean asynchrony ($e_k - \text{mean}_k(e_k)$), not the metronome beat. In the population analysis (Fig. 2a–c), we calculated the correlation in each group using data from all subjects together. In the single-subject analysis (Fig. 2d), we used data from both blocks.

## Autoregressive model (Supplementary Note 2 and Supplementary Fig. 4)

To study the timescale of serial dependence in tapping tasks, we used an autoregressive model, where each asynchrony is predicted by several previous asynchronies (with no intercepts, since we used the perceived asynchronies ($e_k - \text{mean}_k(e_k)$). To determine the number of previous asynchronies to use in the model, we ran a stepwise regression analysis both at the group level and for each participant separately. In each step of the regression, an additional asynchrony (going one tap back from the earliest asynchrony already incorporated into the model) was added if the $F$ value of the SSE (sum of square errors) had a $p$ value < 0.1. In the group model, we used separate predictors for each subject, but the $F$ value was calculated based on adding an additional predictor for all subjects in the group. The final group-level model included three predictors in all three groups, and at the single-participant level, it included one to three predictors for 103/109 participants, indicating that phase correction is a rapid process. We, therefore, fit an autoregressive model with four predictors (for all participants, we tried to predict asynchrony $k$ with asynchronies $k$-1, $k$-2, $k$-3, and $k$-4). Formally, our model can be written as:

$$e_k = b_1 e_{k-1} + b_2 e_{k-2} + b_3 e_{k-3} + b_4 e_{k-4} + \xi_k \tag{5}$$

where $\xi_k$ is independent Gaussian noise. The model combined data from both experiment blocks.

## Computational model of sensorimotor synchronization

To test whether individuals with autism show noisier representations or "sloppier" motor production we used a computational model of sensorimotor synchronization[29,31,32]. The model assumes that the interval between two consecutive taps is influenced by three components: timekeeping of the metronome tempo, motor execution, and phase correction. Formally, the model can be written as follows (see Fig. 3a):

$$r_k = -\alpha e_{k-1} + T_k + M_k - M_{k-1} \tag{6}$$

Where $r_k$ is the time interval between the participant's $k$-1 and $k$ taps and $e_{k-1}$ is the perceived asynchrony at beat $k$-1 (relative to the mean asynchrony). $T_k$ is the representation of the metronome tempo (timekeeper), which is composed of two parts—a fixed mean ($t_0$) and a Gaussian noise component ($n_k$), assumed to have zero mean and variance $\sigma_T^2$ ($\sigma_T$ is referred to as timekeeper noise). $M_k$ models the noise in the motor processing also assumed to be Gaussian with zero mean and variance $\sigma_M^2$ ($\sigma_M$ is referred to as motor noise). Lastly, we denote by $\alpha$ the phase correction, which is the proportion of the previously perceived asynchrony that is corrected in the next tap. Optimally, positive asynchrony deviations should be followed by shorter intervals, therefore the phase correction parameter $\alpha$ appears with a negative sign. This way $\alpha = 1$ corresponds to full correction, and similarly, $\alpha = 0$ will mean that the participant's asynchrony is carried fully into the next tap. The contribution of the timekeeper and motor noise to performance can be separated since they influence the covariance of inter-tap-intervals differently—only the motor noise influences both $r_k$ and $r_{k-1}$. Ref. [28] showed that a naïve implementation of this approach results in biased estimates, but under the assumption of an upper bound on the magnitude of the motor noise ($\sigma_M < \sigma_T$), the parameters of the model can be reliably estimated.

We fit the model for each block separately and averaged the two repetitions of the isochronous condition (Fig. 3). Blocks with more than 40% missing values (omitted or excluded taps) were excluded from this analysis (three blocks altogether, two from the same subject which was excluded from the computational model results). Parameter fit was performed using the bGLS method described in ref. [28]. Importantly, the version of the algorithm for parameter extraction in ref. [28] does not enable fitting with missing values. We adapted the algorithm to enable fitting missing data (Supplementary Note 5). Adequate parameter recovery using this method is shown in Supplementary Note 3 and Supplementary Fig. 6.

*Experiment 2*

## Response dynamics to changes in tempo

To assess how participants respond to changes in the tempo we aligned the participants' responses to the tempo change and averaged each participant's responses to acceleration and deceleration separately (Fig. 4). For presentation purposes, we aligned the baseline delay intervals to the metronome tempo by subtracting the average asynchrony in the two intervals before the change from the delay interval values of the entire segment. We

included only transitions where all responses were available from two taps before the change (to establish a baseline asynchrony) to seven taps after the change (to assess the full progression of the adaptation procedure). Transitions with missing values in this range, or that were too close to the start or end of the block were excluded. Figure 4 shows only participants with at least two repetitions of a given transition magnitude and direction (for each step-size and transition direction between one-six participants were excluded across all groups).

## Update to changes after several taps

We used the distributions of the delay intervals under each metronome tempo separately (using data from both repetitions of each condition). We excluded the four beats immediately following the change (including the moment of change; taking out two–six beats after the change produced similar results). If participants fully adapt to the change, the two distributions should be highly separable. We quantified this using three measures (section Individuals with ASD do not fully update to tempo changes even following several seconds, Fig. 5):

1. Sensitivity index, or d′: the difference between the means normalized by the pooled SDs:

$$d' = \frac{\mu_{d_1} - \mu_{d_2}}{\sqrt{(\sigma_{d_1}^2 + \sigma_{d_2}^2)/2}} \tag{7}$$

Where $\mu_{d_1}$ and $\mu_{d_2}$ are the means of distributions 1 and 2 and $\sigma_{d_1}^2$ and $\sigma_{d_2}^2$ are the variances.

2. AUC: we create a ROC curve by varying the threshold of a binary classifier designed to discriminate between the two distributions (such that a delay interval below the threshold is marked as short tempo, and a delay interval above the threshold is marked as long tempo). For each threshold, we calculate the percentage of true positives (TPR true positive rate, delay intervals in the short tempo that were classified correctly) and false positives (FPR false positive rate, delay intervals in the long tempo that were classified incorrectly as short tempo). AUC is defined as:

$$AUC = \int_0^1 TPR(FPR^{-1}(x))dx \tag{8}$$

3. Difference between the means of the distributions (without normalizing):

$$\mu_{d_1} - \mu_{d_2} \tag{9}$$

Where $\mu_{d_1}$ and $\mu_{d_2}$ are the means of distributions 1 and 2.

## Extended computational model of sensorimotor synchronization

To understand whether individuals with autism manifest an impairment in their response to external changes, we used an extension to the computational model of Experiment 1 proposed by ref. [42], by enabling the mean of the timekeeper to vary in each interval, i.e.,

$$T_k = t_k + n_k \tag{10}$$

The mean $t_k$ is expected to dynamically track the changes in tempo. This is implemented by adding the following dynamics:

$$t_k = t_{k-1} - \beta e_{k-1} \tag{11}$$

Where $\beta$ denotes the period correction rate, which is the proportion of the previous asynchrony corrected in each interval. When the tempo suddenly gets slower (deceleration), this will create a large negative asynchrony, since the participant will tap too early, expecting the metronome at the time of the previous tempo. This change requires the internal period estimate to be elongated, and since the asynchrony, in this case, is negative $\beta$ (the period correction parameter) appears with a negative sign. The full model is defined by the coupled equations (Eqs. (2) and (7)), substituting Eq. (6) (see Figs. 3a and 6a):

$$r_k = -\alpha e_{k-1} + t_k + n_k + M_k - M_{k-1}$$
$$t_k = t_{k-1} - \beta e_{k-1} \tag{12}$$

To combine these into one equation and fit the model we use the difference between the model equation at time $k$ and at time $k$-1:

$$r_k - r_{k-1} = -\alpha e_{k-1} + \alpha e_{k-2} + T_k - T_{k-1} + M_k - 2M_{k-1} + M_{k-2} \tag{13}$$

Note that:

$$T_k - T_{k-1} = t_k - t_{k-1} + n_k - n_{k-1} = -\beta e_{k-1} + n_k - n_{k-1} \tag{14}$$

So from Eqs. (13) and (14) we get:

$$r_k - r_{k-1} = -(\alpha + \beta)e_{k-1} + \alpha e_{k-2} + n_k - n_{k-1} + M_k - 2M_{k-1} + M_{k-2} \tag{15}$$

As in Experiment 1, the covariance structure can be used to disentangle the noise terms (although the specific structure is different, see Appendix of ref. [28]).

To enhance the model's sensitivity to changes, we fit the model separately for each tempo change segment (from two beats before the change to seven beats following the change, see section Response dynamics to changes in tempo) and average the resulting model

estimates within each block. Importantly, the mean asynchrony (needed to adjust the asynchronies relative to the participant's perception) are estimated based on the entire block[28], therefore we excluded blocks with >40% missing values, as in Experiment 1. This led us to exclude eight blocks altogether: one block from 50 ms step-size, four blocks from the 70 ms step-size, and three blocks from the 90 ms step-size. Within the remaining blocks, we excluded segments with missing values, as in section Response dynamics to changes in tempo. Adequate parameter recovery using this fitting method (including the split into tempo change segments) is shown in Supplementary Note 3 and Supplementary Fig. 6.

## Model comparison

The extended computational model was compared to a model without timekeeper dynamics, that is, a model defined according to Eq. (15), with period correction ($\beta$) set to zero:

$$r_k - r_{k-1} = -\alpha(e_{k-1} - e_{k-2}) + n_k - n_{k-1} + M_k - 2M_{k-1} + M_{k-2} \qquad (16)$$

The models were compared for each subject separately, using the likelihood ratio test and AIC.

## Combined measures

For the model-free (Fig. 5) and model-based analyses (Fig. 6), combined measures were calculated by z-scoring each step-size separately, then averaging over the different step-sizes. This was done to account for the different scales of parameters estimated using different step-sizes. Z-scoring was performed using the mean and SD of the neurotypical group. Similarly, the combined update rate (Fig. 7) was formed by z-scoring the phase correction estimate from Experiment 1 ($\alpha$), and the combined period correction estimate from Experiment 2 ($\beta$) and averaging them.

**AQ50 questionnaire**. Nineteen of 30 participants with ASD and 37 of 47 neurotypical participants completed the AQ50 questionnaire[44]. None of the participants with dyslexia were asked to fill the AQ50. AQ50 questionnaire data were not acquired for all neurotypical and ASD participants since it was added only after we began collecting other experimental data.

The AQ50 is a self-report questionnaire, aimed to evaluate the presence of several traits which are characteristic of individuals with ASD, both in ASD and in neurotypical populations. It was recently shown that some questions in the AQ50 differentially bias neurotypical and individuals with ASD[79], therefore we used the three-factor model of the AQ50 proposed by ref. [45], which is less influenced by these biases. We compared our calculated update rate to the social skills factor and the communication/mindreading factor.

(*) indicate reverse keyed items. Responses vary from 0 (definitely disagree) to 3 (definitely agree).

The items in the social skill factor are:

1. I am good at social chit-chat*.
2. I find social situations easy*.
3. I enjoy social occasions*.
4. I enjoy social chit-chat*.
5. I frequently find that I do not know how to keep a conversation going.
6. I enjoy meeting new people*.
7. I find it hard to make new friends.
8. When I was young, I used to enjoy playing games involving pretending with other children*.
9. I find myself drawn more strongly to people than to things*.
10. I enjoy doing things spontaneously*.
11. I find it very easy to play games with children that involve pretending*.
12. I would rather go to a library than to a party.

Notably, a large proportion of items in this factor (4/12) begin with the words "I enjoy", indicating a tendency to enjoy social situations, but not necessarily social skills. Individuals with autism often crave social situations, despite being judged as poor performers in this respect[2].

The items in the communication/mindreading factor are:

1. People often tell me I keep going on and on about the same thing.
2. When I am reading a story, I find it difficult to work out the characters' intentions.
3. I find it difficult to work out people's intentions.
4. I am often the last to understand the point of a joke.
5. Other people frequently tell me that what I have said is impolite, even though I think it is polite.
6. If there is an interruption, I can switch back to what I was doing very quickly*.

To determine whether we can combine the two groups (neurotypical and autism) to calculate the correlation between update rate and responses on the communication subscale we performed a bootstrap permutation analysis, designed to show that the correlation values in the two groups can be approximated using the correlation in the combined sample, that is, that the correlations in the two groups are not significantly different than the combined correlation, or different from each other. To do this, we created surrogate distributions by resampling (with

replacement) data from the two participant groups separately, so that we formed one distribution for neurotypical values and another for autism values. We then calculated the Spearman correlations in each group separately, and on the combined sample, and calculated the differences between these correlations. This procedure was repeated 1000 times. Finally, we compared the resulting distributions of differences between correlation values to those of the experimental data, and in all cases the difference in the experimental data is well within the null distribution (all $p > 0.5$).

**Reporting Summary**. Further information on research design is available in the Nature Research Reporting Summary linked to this article.

## Data availability

All data generated in this study have been deposited in the OSF public repository (https://doi.org/10.17605/OSF.IO/83WNU)[80]. Source data are provided with this paper.

## Code availability

The custom code used to analyze the data in this study (including the implementation used for the bGLS algorithm) and create all figures (except Figs. 1a, 3a, and 6a) is publicly available at Zenodo (https://doi.org/10.5281/zenodo.4930034)[81]. Source data are provided with this paper.

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

## Acknowledgements

We thank O. Guri and S. Granot for their help collecting experimental data. We thank H. Vishne for assistance with the figures. We thank P. Dayan, U. Frith, and Y. Hart for their support and helpful comments on the manuscript. This project has received funding from the European Research Council (ERC) under the European Union's Horizon 2020 research and innovation program (grant agreement No 833694) and the Israel Science Foundation (Grant No. 1650/17), both awarded to Merav Ahissar.

## Author contributions

Conceptualization: G.V., N.J. and M.A. Formal analysis and methodology: G.V. and N.J. Writing—original draft: G.V. and M.A. Writing—review and editing: G.V., N.J. and M.A. N.J. designed the experiments. T.M. initiated the collection of ASD data. G.V., O.F., T.M. and T.E. collected the data. Funding acquisition and supervision: M.A.

## Competing interests

The authors declare no competing interests.
