## [Peer Review File · Nature Communications]

Slow update of internal representations impedes synchronization in autismREVIEWER COMMENTS

Reviewer #1 (Remarks to the Author):

Using a paced finger tapping task, Vishne et al show that recent asynchronies ("prediction errors") influence current tap timing less in autism (less error correction), and that when the metronome tempo changed, participants with autism were also slower in adapting to the new tempo. They interpret this as consistent with their own published findings (and theory) about slower prior updating in ASD.

It is nice to see the authors' earlier theory and evidence (Lieder et al Nature Neuroscience) confirmed in a completely different task context. It illustrates again that the 'slow update' hypothesis is a serious contender among the candidate Bayesian explanations of autism. I found the study very interesting, methodologically sound, and theoretically informative, but I have concerns about both of the central claims (from the title/abstract). I discuss motor plan updating first and the link with social skills next.

- We learn on line 751 that "The model takes as an input the empirical d_k and e_k ...". Of course e_k (asynchrony at k) is not an empirical given from the subject's point of view. It's a (subjectively computed) perceptual inference of the discrepancy between an auditory signal and a proprioceptive signal (associated with one's own tap). Possible differences in this perceptual inference are clouded by the model used (they will end up in the update weight α or β). If one considers the findings on simultaneity and order judgments in ASD and the wider temporal binding/integration window (for multisensory stimuli; eg see Stevenson et al Journal of Neuroscience, 2014; Foss-Feig et al 2010; see review Wallace & Stevenson 2014, Neuropsychologia), it is extremely plausible that differences already emerge in this perceptual inference/integration process. Those differences would precede those focused on the current paper (differences in "updating of motor plans"), but, as far as I can see, they would be indistinguishable from the latter in the current data/task. In other words, the perceived asynchrony (error) could be smaller in ASD (because of reduced temporal resolution) leading to similar alterations to the reported ones. Incidentally, such a perceptual origin of differences also explains why unpaced tapping is intact (as described in the Discussion).

- The second main claim is that slowness in phase/period correction is linked to social problems in autism. We lack some data to properly evaluate this claim. The reported significant negative correlation between the social skills subscale of the AQ is of course a necessary condition. And in general, downstream effects (including on social skills) of altered sensorimotor processes (such as identified by the authors) are very plausible. However, it would also be necessary to have the correlations with the other subscales, as well as the correlations within the other groups (only data for ASD group reported, and only for a subset of participants), to judge the specificity of the claim. To make this the main claim (in title and abstract), suggesting that slow updating is the origin of poor social functioning in autism, I think more information is needed here. Especially given that, 1) diagnosis was not confirmed using a standardized tool, 2) there is a tendency towards decreased period correction in dyslexia as well (specifically, their period correction does not significantly differ from ASD), a group in which, presumably, social skills would be typical. It would seem to be important to have AQ data (and correlations with correction weights) for this group and the neurotypical group, as well. Based on the results, the authors can adjust their claims about the relationships between the slower update and the social skills.

- I had a few secondary comments as well. It would be interesting if the authors could discuss a bit more the fact that there was no difference in mean asynchrony between ASD and NT or DYS participants, despite a difference in dynamics. What does this mean in terms of underlying learning mechanisms?

- Further, one of the authors' claims is that there are core differences in the learning dynamics in each group. I wondered whether they have assessed whether there is an improvement across blocks. In particular, in experiment 2, participants may have noticed the underlying structure of the environment (change of tempo every 8 to 12 trials, and alternating increase and decrease), which may have helped them anticipate and adapt to it.

Finally a few minor points:

- Figure 1: Is there a correlation between the mean asynchrony and the SD (which would suggest that the ASD outliers are the same participants in the b and c plots)?

- Typo: p.14: verb missing after 'post-hoc comparisons'; also on line 39: "leading" is missing "to"

- What was the percentage of excluded trials in each experiment and group? It is also written that blocks with more than 40% missing values were excluded. How many blocks were excluded on average per participant in each group?

In sum, these remarks certainly do not invalidate the authors' findings: The study is well-designed and analyzed, and the results add to the authors' previous findings/theory, with a new task, rather large sample sizes, and the advantage of having both an ASD and a DYS group. However, these comments do urge caution in interpreting the findings in the way the authors do. Specifically, the claims in the title and abstract (and main text) are too strong given the results (i.e., it would only be fair to say that people with ASD are slower to adapt their tempo, rather than mentioning motor plans and social skills). Due to these interpretational difficulties, the study may be perceived as more incremental than their previous (Lieder et al) one, but I imagine the authors may be able to tone down claims and/or substantiate them more (at least for claim 2) than is currently the case.

Best wishes,

Sander Van de Cruys

Reviewer #2 (Remarks to the Author):

This study tests current Bayesian theories of ASD and particularly whether ASD is related to an overestimation of volatility or a slow update of priors. The literature is currently confusing and as such, I think this study is timely, important and of interest to a wide audience.

The study uses a paced finger tapping task where participants are asked to synchronise tapping to the beat of a metronome. In experiment 1, the beat remains the same, while it changes periodically in experiment 2. The authors test a group of 30 ASD participants which they compare to a matched group with dyslexia (32) and another NT (47).

In experiment 1 they find that the mean asynchrony is similar across groups but the variability is larger in ASD. Computation of correlations between consecutive asynchronies and an autoregressive model show that this is due to reduced online error correction in ASD.

Moreover, a computational model of sensorimotor synchronisation shows that the difference is not due to elevated noise level in either motor or tempo keeping processes.

Experiment 2 has a tempo change. The authors find that individuals with ASD adapt only partially after a change. Modelling suggests that this is due to a slow-period updating in ASD and that this impairment is correlated with the impaired error correction in experiment 1 suggesting it is part of the same mechanism.

The authors finally propose that fast sensorimotor integration is impaired in ASD due to slow updating of internal models.

I found the task to be very interesting, the paper to be very clear and thorough in its methodology, first presenting convincing raw data/model-free differences before dissecting them using model-based analysis. The figures are clear and convincing too. The paper clearly adds to the current findings and discussions in the field.

I have the following comments, however. I think that addressing them would significantly strengthen the paper:

- 1) I would appreciate more description of the ASD group. What diagnostic tools were used for recruitment? ADOS? Would it be possible to see the distribution of AQ scores? How do they compare with the general population?
- 2) Would it possible for the authors to show that adequate parameter recovery can be obtained for both computational models? (see e.g. Wilson & Collins, eLife 2019). ^[1]_{SEP}
- 3) Similarly, it would be reassuring if the authors could show that the models satisfactorily account for the critical features of the data and group differences.
- 4) In the main text, it is not clear how T_k and alpha and beta enter the full model description. I see the description is in supplementary information but I think it would help to have the full equation in the main text.
- 5) More generally, the model and how it is derived could be better explained. An explanatory figure could possibly help getting a more intuitive understanding of the elements of the model.
- 6) The authors mention that there are significant correlations between AQ_social_subscore and the rate of update. Does that mean that the rate of update is not correlated with the other sub-scales then? with global AQ scores? It would help to be precise about this. It is also unclear why only 19/30 ASD participants took the AQ?
- 7) Since the authors frame the paper in the context of current Bayesian theories, it is somewhat surprising that the model that is used is not Bayesian in nature. It would greatly improve the paper if the authors could relate their models to Bayesian/predictive coding models and/or offer a possible Bayesian interpretation of their task/findings. ^[1]_{SEP}
- 8) line 661: "participants. The final group consisted of 109 participants (47 neurotypical, 30 females; 32 dyslexia)." I imagine you mean autism not "females".

Peggy Seriès.

Reviewer #3 (Remarks to the Author):

The present studies required autistic, dyslexic and neurotypical participants to synchronise finger taps with an auditory metronome. In Experiment 1 the metronome was always presented at 2 Hz. In Experiment 2 the metronome accelerated or decelerated – switching between the two every 8-12 trials. All participants tapped before the beat, replicating previous findings. Experiment 1 demonstrated no difference in absolute error between groups, but greater within-participant variability in the autism group. They also found that temporal error on trial N-1 predicted temporal error on trial N to a greater extent in the autism group, indicating that they are insufficiently using sensorimotor error to update actions. Experiment 2 demonstrated that the autism group were slow to adapt to the new tempo relative to the other two groups. The rate of update across both experiments also correlated with social score on the AQ50.

The findings are interesting and I agree that they provide support for the slow updating model of autism. However, I was unsure how the claims fitted with some previous findings, as well as how much advance the present data make over previous studies.

Advance with respect to Lieder et al. 2019. Some of the present authors showed, in this previous study, a biasing effect that was consistent with the slow updating model of autism. While the present

task is different it was not made clear to the reader what conceptual advance it made over and above this previous study. That is, the main take home message is that the data are consistent with the slow updating account of autism. It would strengthen the manuscript to focus on the particular theoretical advance made by these particular data – what do these data allow us to conclude about model updating in autism that could not be concluded from the study last year? What does it add mechanistically to conclusions that this is a motor task? I would expect that conclusion to represent the focus in the abstract, for example.

The authors conclude throughout their manuscript that the autism group demonstrate “poor sensorimotor synchronisation”. This seems a strange conclusion to me given that there was no difference in the mean asynchrony in Experiment 1. Greater variability does not seem to indicate poorer synchronisation. I.e., this result means that trials where they are slow to learn from error are being offset by other trials where they are especially accurate. Any idea why this mean difference is not observed? I am especially interested in how this also would link with the group difference in mean bias in the Lieder et al. (2019) because, given the common proposed underlying mechanism, I would have expected a mean group difference also in this study on the basis of those previous data.

Relatedly, another study demonstrates equivalent sensorimotor synchronisation in autism in a finger tapping task to metronome and should be incorporated. Edey et al. (2018; *J Autism Developmental Disorders*) found comparable absolute error between neurotypical and autistic groups when synchronising to an auditory metronome. However, they also found *lower* absolute error when synchronising to a visual metronome. How would the authors explain this difference between sensory modalities in interpreting their data? The Edey task included a different frequency metronome on each trial of four events, so if autistic individuals are slower to update models why do they show superior accuracy in this study?

Minor comments:

1. The study excludes many participants on the basis of certain scores on the WAIS, absolute reading scores, or ‘extreme’ asynchrony. Were these decisions pre-registered anywhere? It would seem more justifiable to do this according to standard deviations away from the mean.
2. When the authors describe the slow updating model in the introduction, they say that Bayesian updating may be “adequate”, yet the rate of updating of priors is “slow”. This sounds inadequate, rather than adequate, to me. I would therefore suggest the phrasing should be altered.
3. In the regression with four time intervals included as regressors, I would assume that there would be a great deal of overlap in the explained variance by these four regressors – especially given the correlations presented earlier. Has this analysis therefore left out most variance from the analysis by adding all regressors together? Given the way this analysis is set up, it may make more sense to conduct a stepwise regression, where step 1 includes t-1, then t-1 and t-2, then t-1, t-2, and t-3 etc. This way you ask whether additional variance is explained with the addition of each new step away from the present trial.
4. I would incorporate into the results (or introduction) that the metronome is auditory. This information seems crucial, given Edey 2018 (above), yet it was only mentioned for the first time in the methods at the end.

We are grateful to the three reviewers for their thoughtful questions and recommendations, which we feel, have allowed us to substantially improve the manuscript. Below we detail a point-by-point reply to the reviewers' comments. The letter includes all the reviewers' comments on the manuscript.

Reviewer #1 (Remarks to the Author):

Using a paced finger tapping task, Vishne et al show that recent asynchronies ("prediction errors") influence current tap timing less in autism (less error correction), and that when the metronome tempo changed, participants with autism were also slower in adapting to the new tempo. They interpret this as consistent with their own published findings (and theory) about slower prior updating in ASD.

It is nice to see the authors' earlier theory and evidence (Lieder et al Nature Neuroscience) confirmed in a completely different task context. It illustrates again that the 'slow update' hypothesis is a serious contender among the candidate Bayesian explanations of autism. I found the study very interesting, methodologically sound, and theoretically informative, but I have concerns about both of the central claims (from the title/abstract). I discuss motor plan updating first and the link with social skills next.

- We learn on line 751 that "The model takes as an input the empirical d_k and e_k ...". Of course e_k (asynchrony at k) is not an empirical given from the subject's point of view. It's a (subjectively computed) perceptual inference of the discrepancy between an auditory signal and a proprioceptive signal (associated with one's own tap). Possible differences in this perceptual inference are clouded by the model used (they will end up in the update weight α or β). If one considers the findings on simultaneity and order judgments in ASD and the wider temporal binding/integration window (for multisensory stimuli; eg see Stevenson et al Journal of Neuroscience, 2014; Foss-Feig et al 2010; see review Wallace & Stevenson 2014, Neuropsychologia), it is extremely plausible that differences already emerge in this perceptual inference/integration process. Those differences would precede those focused on the current paper (differences in "updating of motor plans"), but, as far as I can see, they would be indistinguishable from the latter in the current data/task. In other words, the perceived asynchrony (error) could be smaller in ASD (because of reduced temporal resolution) leading to similar alterations to the reported ones. Incidentally, such a perceptual origin of differences also explains why unpaced tapping is intact (as described in the Discussion).

We agree, the slower processing stage could be at the perceptual level and the motor manifestations could be inherited. Namely, the tapping task requires both fast and accurate error calculation, which requires fast comparisons between the timing of the external metronome and the proprioception of the tap. If cross modal integration is sloppier in autism (Wallace & Stevenson 2014; Steveneson et al., 2014), then perhaps occasionally no error is

calculated leading to underestimation of the error, and reduced synchronization. In our model it would lead to smaller alphas and betas.

This is now discussed in an added paragraph to the Discussion (paragraph beginning with “The specific stage of processing which yields the slow update is difficult to pinpoint”). It also influenced the focus of the following paragraph.

- The second main claim is that slowness in phase/period correction is linked to social problems in autism. We lack some data to properly evaluate this claim. The reported significant negative correlation between the social skills subscale of the AQ is of course a necessary condition. And in general, downstream effects (including on social skills) of altered sensorimotor processes (such as identified by the authors) are very plausible. However, it would also be necessary to have the correlations with the other subscales, as well as the correlations within the other groups (only data for ASD group reported, and only for a subset of participants), to judge the specificity of the claim. To make this the main claim (in title and abstract), suggesting that slow updating is the origin of poor social functioning in autism, I think more information is needed here. Especially given that, 1) diagnosis was not confirmed using a standardized tool, 2) there is a tendency towards decreased period correction in dyslexia as well (specifically, their period correction does not significantly differ from ASD), a group in which, presumably, social skills would be typical. It would seem to be important to have AQ data (and correlations with correction weights) for this group and the neurotypical group, as well. Based on the results, the authors can adjust their claims about the relationships between the slower update and the social skills.

We agree. Our report included only the ASD data, which were not complete since not all ASD participants agreed to fill an AQ50 questionnaire. However, most of our neurotypical participants (37/47) filled the AQ50 questionnaire. These were not included in the original submission, and we now address their data as well. Participants with dyslexia were not asked to fill the AQ questionnaire (in hindsight we agree it was a mistake).

Our exclusion of neurotypical participants' AQ replies in the originally submitted version was aimed to simplify the presentation, since filling the AQ50 is differently biased in neurotypical and ASD individuals (e.g. Agelink van Rentergem et al., 2019). Your comment convinced us to include both populations. To accommodate both populations in a unified framework we used the 3-factor model proposed by Austin (2005), which improved the factor structure originally proposed by Baron-Cohen et al. This is now detailed in a newly added subsection of the Methods (entitled: AQ50 questionnaire and communication skills). This analysis allowed us to ask whether communication skills and rate of updating are correlated in ASD and neurotypical populations, and whether the pattern of correlation is similar. We found that the pattern is similar (quantitatively verified by permutation tests, also described in the same Methods subsection), so we could calculate correlations for the two populations together. These were highly significant. The dependence in each population and in the merged population is presented in the newly added Fig. 8 and the detailed statistics of the correlations is presented in a new section of the Results (entitled: Update rate is correlated with communication and mindreading skills).

Following your suggestion, we have added the neurotypical data, but (regrettably) had not collected dyslexia data. Though we agree that it had not been a good choice - we believe that we have a strong correlation-based case for the two populations that we did measure. Still, we agree that since the data set are not complete, we should tone down the claims regarding correlation. Accordingly, this claim was omitted from the title and moderated in the abstract.

Regarding the confirmation of diagnosis – True – not all are participants were diagnosed with the same tool, but all our participants with ASD were formally diagnosed, and were eligible to government support, due to their ASD diagnosis. We now explain this in the manuscript in the sub-section Participants of the Methods section. Given that neurotypicals (now added) show the same correlation as individuals with ASD (though ASD have poorer error correction and poorer social understanding), we think that this is addressed in the revised manuscript.

Regarding the dyslexia group – In the previous version of the manuscript the combined rate of period correction was not significantly different between the group with autism and the group with dyslexia, as you noted, though the average of dyslexia and control was very similar. Given your comment on our exclusion criteria we refined our criteria (see our detailed answer below) and also adjusted the z-scoring procedure. In the previous version of the manuscript z-scoring was done with respect to the mean and standard deviation of the entire sample. Since special populations often show more variable responses, this normalization is more sensitive to exclusions than normalization with respect to neurotypicals only, which is more common (e.g. Ramus et al., 2003). We therefore refined our normalization, and it is now with respect to the neurotypical group alone. Together they slightly increased the significance of the difference between the group with ASD and the group with dyslexia. In the updated calculation, both period correction (Figure 6b) and the combined rate of update (Figure 7d) are significantly different between the ASD and dyslexia group.

Still, as you note, sensitivity to small tempo changes is marginally smaller in dyslexia (figure 4e-f) than in neurotypicals. This is not due to slow update in dyslexia, since large changes are quickly updated (e.g. figure 4 a-b) and the calculated update rate is similar to neurotypicals'. Based on our previous studies of dyslexia (Banai & Ahissar, 2006) We interpret this as a consequence of their mildly poorer perceptual interval discrimination, which we attribute to their impaired ability to use the repetitions of the stimulus (in line with Lieder et al., 2019), rather than to slow updating. This point was not clarified in the original manuscript and is now explained (added to the text in the Results (see "Since the sluggish update in dyslexia is manifested only in the small tempo change...") and added in the Discussion (last paragraph before conclusion)).

- I had a few secondary comments as well. It would be interesting if the authors could discuss a bit more the fact that there was no difference in mean asynchrony between ASD and NT or DYS participants, despite a difference in dynamics. What does this mean in terms of underlying learning mechanisms?

We have now added the following explanation to the Introduction (third paragraph):

“First, tapping is perceived as synchronous when individuals tap slightly ahead of each beat, hence the perceived synchrony is characterized by a small negative asynchrony (Aschersleben & Prinz, 1995; Repp 2005; Fig. 1a). The mean of this asynchrony is considered to reflect mainly low-level limitations, such as different processing latencies between auditory and somatosensory proprioceptive signals (Repp 2005; Repp & Su, 2013).”

- Further, one of the authors' claims is that there are core differences in the learning dynamics in each group. I wondered whether they have assessed whether there is an improvement across blocks. In particular, in experiment 2, participants may have noticed the underlying structure of the environment (change of tempo every 8 to 12 trials, and alternating increase and decrease), which may have helped them anticipate and adapt to it.

Since all experimental blocks were repeated twice, we were able to compare the first and second assessment on the two key behavioral measures: phase correction (Experiment 1) and period correction (Experiment 2). Pooling across groups we found no significant differences in any of the measures (all $p > 0.2$, Wilcoxon signed rank test), or in the single groups (after controlling for multiple comparisons). Our interpretation is that the basic structure of tempo switching is learned within the first block, since each block consists of many switches, whereas accurate evaluation of the number of beats before tempo switch (if tempo was not switched after 11 isochronous intervals it should switch now) was not learned within a session. Since none of the groups improved, we did not integrate it into the manuscript.

Finally a few minor points:

- Figure 1: Is there a correlation between the mean asynchrony and the SD (which would suggest that the ASD outliers are the same participants in the b and c plots)?

In the neurotypical group there is a moderate, significant negative correlation between mean asynchrony and SD (Spearman correlation $\rho_{CON} = -0.37$, $p = 0.01$), meaning that participants with more extreme deviations in asynchrony (more negative) show larger variability. But the correlations were not significant in either the dyslexia or the autism groups ($\rho_{DYS} = -0.24$ ($p = 0.18$) and $\rho_{ASD} = -0.2$ ($p = 0.3$)). Of the six participants with autism who had the most extreme SD values, only one had an extreme (below -60ms) mean asynchrony, i.e. the outliers are largely separate. This point is now clarified in the Results (legend of Figure 1).

- Typo: p.14: verb missing after 'post-hoc comparisons; also on line 39: "leading" is missing "to"

Corrected in both sentences.

- What was the percentage of excluded trials in each experiment and group? It is also written that blocks with more than 40% missing values were excluded. How many blocks were excluded on average per participant in each group?

Thanks – The detailed exclusion numbers and criteria for both experiments was added to the Methods section (subsection finger tapping analysis), and is also described below:

Overall, there was a small number of omitted or excluded taps (cases where the participant did not tap within a 400ms window around the metronome beat) - less than 5% of the taps (across experiments). In Experiment 1 the percentage of omitted or excluded taps across groups was: median [interquartile range] (%): neurotypical: 0.5 [1.3], dyslexia: 0.5 [1.5], autism: 1.2 [4.9]. The difference between the groups was marginally significant (Kruskal Wallis test, $H(2)=5.8$, $p=0.055$), corresponding to our finding of more variable tapping in the autism group. In Experiment 2 the percentages were (median [interquartile range] (%)): neurotypical: 1.3 [2.5], dyslexia: 1.5 [3], autism: 2.8 [10], which is again marginally significant (Kruskal Wallis test, $H(2)=5.58$, $p=0.06$). Exclusion was performed in accordance with previous literature (Repp, 2005), and was done only when the taps could not be assigned with certainty to a specific metronome beat. Note that excluding these taps only decreases the effects we found, since these taps are characteristic of highly variable tapping, and excluding them results in lower standard deviations. Exclusion of entire blocks was done only in the computational modelling analyses, for blocks with more than 40% omitted or excluded taps where the model could not be reliably estimated. We excluded only a small number of blocks: Experiment 1 - 3 blocks out of 218 blocks from all subjects, all from the ASD group; Experiment 2 - 8 blocks out of 654 blocks from all subjects and conditions, two from the neurotypical group and six from the autism group. This exclusion too is decreasing our group difference effect, since the most variable blocks are excluded.

Given this question, we slightly changed the exclusion criteria in Experiment 2 to match those of Experiment 1. Initially exclusion was based only on missing steps within the segments analyzed for tempo change. We now exclude from Experiment 2 all blocks with > 40% missing taps, as explained above. The effect of this change to model calculations is small.

In sum, these remarks certainly do not invalidate the authors' findings: The study is well-designed and analyzed, and the results add to the authors' previous findings/theory, with a new task, rather large sample sizes, and the advantage of having both an ASD and a DYS group. However, these comments do urge caution in interpreting the findings in the way the authors do. Specifically, the claims in the title and abstract (and main text) are too strong given the results (i.e., it would only be fair to say that people with ASD are slower to adapt their tempo, rather than mentioning motor plans and social skills). Due to these interpretational difficulties, the study may be perceived as more incremental than their previous (Lieder et al) one, but I imagine the authors may be able to tone down claims and/or substantiate them more (at least for claim 2) than is currently the case.

Best wishes,

Sander Van de Cruys

Dear Sander Van de Cruys, thank you for your insightful comments. Incorporating them into our manuscript has helped us present it in a broader perspective, and improved it.

Reviewer #2 (Remarks to the Author):

This study tests current Bayesian theories of ASD and particularly whether ASD is related to an overestimation of volatility or a slow update of priors. The literature is currently confusing and as such, I think this study is timely, important and of interest to a wide audience.

The study uses a paced finger tapping task where participants are asked to synchronise tapping to the beat of a metronome. In experiment 1, the beat remains the same, while it changes periodically in experiment 2. The authors test a group of 30 ASD participants which they compare to a matched group with dyslexia (32) and another NT (47).

In experiment 1 they find that the mean asynchrony is similar across groups but the variability is larger in ASD. Computation of correlations between consecutive asynchronies and an autoregressive model show that this is due to reduced online error correction in ASD.

Moreover, a computational model of sensorimotor synchronisation shows that the difference is not due to elevated noise level in either motor or tempo keeping processes.

Experiment 2 has a tempo change. The authors find that individuals with ASD adapt only partially after a change. Modelling suggests that this is due to a slow-period updating in ASD and that this impairment is correlated with the impaired error correction in experiment 1 suggesting it is part of the same mechanism.

The authors finally propose that fast sensorimotor integration is impaired in ASD due to slow updating of internal models.

I found the task to be very interesting, the paper to be very clear and thorough in its methodology, first presenting convincing raw data/model-free differences before dissecting them using model-based analysis. The figures are clear and convincing too. The paper clearly adds to the current findings and discussions in the field.

I have the following comments, however. I think that addressing them would significantly strengthen the paper:

1) I would appreciate more description of the ASD group. What diagnostic tools were used for recruitment? ADOS? Would it be possible to see the distribution of AQ scores? How do they compare with the general population?

All individuals with autism were diagnosed with autism spectrum disorder by expert clinicians, which is required for government support, to which all were entitled (now added to Participants section of the Methods). Some but not all clinicians use ADOS as part of this process, and hence we did not discuss it.

AQ50 separated between neurotypicals and individuals with autism. As expected from the literature we found higher scores in the autism group (median [interquartile range]: 75 [23.5]) compared with the neurotypical group (median [interquartile range]: 52 [17.5]), and this group difference was highly significant (Wilcoxon rank sum test, $p < 0.0001$, Cliff's delta = 0.68). This information was now added to the Results in a newly added section "Update rate is correlated with communication and mindreading skills".

2) Would it possible for the authors to show that adequate parameter recovery can be obtained for both computational models? (see e.g. Wilson & Collins, eLife 2019). 
Thank you for this important comment. We performed parameter recovery analysis and found very good correspondence between the values used to generate the simulations (the fitted values of the raw data) and the recovered values. In Experiment 1 Spearman correlations between the simulation values and recovered values were above $\rho = 0.92$ for all parameters in all three groups. In Experiment 2 the correlations were above $\rho = 0.91$ for the error correction parameters and the timekeeper parameter in all step-sizes and groups. Motor noise correlations were lower, between 0.55 and 0.83, but RMSE (root mean square error) values were still very low, below 5.1ms in all conditions and groups, namely the model recovered the small magnitude of the motor noise, but because the value itself is so close to zero the estimate is somewhat noisy.

Correlations were even higher for the parameters which showed group differences in our study (phase correction (α) in Experiment 1, and period correction (β) in Experiment 2) – phase correction correlation was above $\rho = 0.94$ in each of the groups and period correction correlations were above $\rho = 0.97$ in all conditions and groups and when calculating the combined z-scored values. This information was added to the supplementary material (section *Computational models parameter recovery*) as Fig. S6, which is also provided below:

Figure S6: Parameter recovery of the error correction parameters from both experiments. (a) Experiment 1 phase correction (α), (b) Experiment 2 phase correction (z-score combined value across blocks), (c) Experiment 2 period correction (z-score combined value across blocks) and (d) Combined update rate (averaging phase and period correction). All error correction parameters showed remarkable similarity between the values used for the simulation (which were the fitted values of each subject), and the recovered values. Each dot represents the average value of one simulation setting (corresponding to one subject), calculated using 1000 simulations. The dashed line represents equality between the recovered and the fitted values. All points are remarkably close to the equality line. Error correction parameters in specific conditions, and the timekeeper parameter all showed similar patterns. Motor noise recovery was noisier in Experiment 2, but nonetheless very close to the values used to generate the data (see values in the supplementary material text).

3) Similarly, it would be reassuring if the authors could show that the models satisfactorily account for the critical features of the data and group differences.

Thank you for this suggestion. We simulated 500 repetitions of the entire experiment based on the fitted parameter values in both computational models. In Experiment 1 group differences in the correlations of consecutive asynchronies were reproduced by the simulated data, albeit with lower effect size than the one observed in the experiment (the effect size was not significantly different), and the simulated values are highly correlated with the values based on the experimental data (Spearman correlation in all groups was larger than 0.9). In Experiment 2 the model reproduced very closely the dynamics of adaptation to changes seen in the different populations, namely, simulations based on the fitted values of the neurotypical and dyslexia groups show quick correction for changes in tempo (especially for the prominent changes in the 70ms and 90ms step sizes). Furthermore, the simulations based on autism data show only partial updating to the change and continue to show a sizeable difference in asynchrony even 7 taps after the tempo change. This information was added to the supplementary material (section *The model reproduces tapping profiles of the different groups*) as Figures S7 (Experiment 1) and S8 (Experiment 2), and is also provided in the figures below:

Figure 4 (Experimental results, left) and Supplementary Figure 8 (Simulated results, right). The computational model reproduces the key features of the experimental data. Namely, individuals with autism adapt to changes in tempo only partially, even when changes are very salient, whereas the neurotypical and dyslexia groups are quick to correct for these changes. This difference is less pronounced for the 50ms step-size, as in the experimental data. (a-b) 90ms step-size, (c-d) 70ms step-size and (e-f) 50ms step-size. In each panel the x-axis represents the metronome-beat number around the moment of tempo change (beat 0), and the y-axis measures the observed delay interval (left) or simulated delay interval (right) in each beat. Error bars denote SEM across experimental data (left) or simulations (right) of different participants.

4) In the main text, it is not clear how T_k and α and β enter the full model description. I see the description is in supplementary information but I think it would help to have the full equation in the main text.

Thank-you, we updated the main text to include a more detailed explanation of the extended computational model.

5) More generally, the model and how it is derived could be better explained. An explanatory figure could possibly help getting a more intuitive understanding of the elements of the model.

We designed two new figures to explain the isochronous computational model (newly added Fig. 3a) and the tempo change dynamics (newly added Fig. 6a), respectively (also added below). The figures are now integrated in a more detailed explanation of the model in the main text.

Figure 3a. A schematic illustration of the computational model used to dissociate error correction mechanisms from poor timekeeping or elevated motor noise (Wing and Kristofferson 1973; Vorberg and Wing 1996; Vorberg and Schulze 2002). Each tapping interval (blue empty arrow) is assumed to be the summation of three mechanisms: (1) error correction based on the previous asynchrony (marked in red, the magnitude of the correction is determined by the phase correction parameter α) (2) timekeeping of the base tempo T_k (composed of a fixed t_0 , purple, plus the noise at tap k , n_k , green) and (3) motor noise (turquoise).

Figure 6a. Illustration of the dynamics added to the isochronous computational model (Figure 3a), which enables tracking of changing tempos (Schulze et al., 2005). The internal period estimate (t_k , purple) is adjusted in each trial based on the recent asynchrony (in red, the magnitude of the correction is determined by the period correction parameter β). To produce the tapping interval, noise is added to this estimate (n_k , green) as in the isochronous model.

6) The authors mention that there are significant correlations between AQ_social_subscore and the rate of update. Does that mean that the rate of update is not correlated with the other sub-scales then? with global AQ scores? It would help to be precise about this. It is also unclear why only 19/30 ASD participants took the AQ?

Thank you. This is now explained in more details and with additional data. In the originally submitted manuscript we did not include neurotypical data (37/47), since neurotypicals and individuals with ASD have different biases when filling AQ50 (e.g. Agelink van Rentergem et al.,

2019). To include both populations and compare the pattern of correlations we introduced Austin's (2005) factors, which do not include the most differentially biased questions. The Austin (2005) factors are detailed in the added AQ50 section of the Methods, including the specific questions composing the communication/mindreading factor that we use.

Importantly, the expected communication/mindreading factor was significantly correlated with the rate of update within each population separately (neurotypicals and ASD). Since the two populations showed similar correlations within each population separately, we also grouped them together (newly added Figure 8). None of the other two factors was correlated with update rate in either population. The precise numbers and plots were added to the new Results section on rate of update and communication.

7) Since the authors frame the paper in the context of current Bayesian theories, it is somewhat surprising that the model that is used is not Bayesian in nature. It would greatly improve the paper if the authors could relate their models to Bayesian/predictive coding models and/or offer a possible Bayesian interpretation of their task/findings.

We have re-written the first two paragraphs of the Discussion, which now specifically relate our modelling to Bayesian models. The relation to Bayesian accounts is now integrated into the second paragraph.

8) line 661: "participants. The final group consisted of 109 participants (47 neurotypical, 30 females; 32 dyslexia)." I imagine you mean autism not "females".

Corrected.

Peggy Seriès.

Dear Peggy Seriès, thank you for your constructive comments. Addressing them had both sharpened our manuscript, and helped us clarify our modelling sections.

Reviewer #3 (Remarks to the Author):

The present studies required autistic, dyslexic and neurotypical participants to synchronise finger taps with an auditory metronome. In Experiment 1 the metronome was always presented at 2 Hz. In Experiment 2 the metronome accelerated or decelerated – switching between the two every 8-12 trials. All participants tapped before the beat, replicating previous findings. Experiment 1 demonstrated no difference in absolute error between groups, but greater within-participant variability in the autism group. They also found that

temporal error on trial N-1 predicted temporal error on trial N to a greater extent in the autism group, indicating that they are insufficiently using sensorimotor error to update actions. Experiment 2 demonstrated that the autism group were slow to adapt to the new tempo relative to the other two groups. The rate of update across both experiments also correlated with social score on the AQ50.

The findings are interesting and I agree that they provide support for the slow updating model of autism. However, I was unsure how the claims fitted with some previous findings, as well as how much advance the present data make over previous studies.

Advance with respect to Lieder et al. 2019. Some of the present authors showed, in this previous study, a biasing effect that was consistent with the slow updating model of autism. While the present task is different it was not made clear to the reader what conceptual advance it made over and above this previous study. That is, the main take home message is that the data are consistent with the slow updating account of autism. It would strengthen the manuscript to focus on the particular theoretical advance made by these particular data – what do these data allow us to conclude about model updating in autism that could not be concluded from the study last year? What does it add mechanistically to conclusions that this is a motor task? I would expect that conclusion to represent the focus in the abstract, for example.

Thank-you for this comment:

First, Lieder et al. proposed the slow-update hypothesis based on a perceptual task. Here we explore the broader relevance – to both motor behavior, known to be sloppy yet not previously explained within (a Bayesian or) any unified conceptual framework, and to social skills.

Second, in the Lieder et al. study there was an incentive (enhanced performance) to under-use previous sensory information, which can hamper performance on the task (bias- trials) when previous frequencies do not provide supportive information. In our sensorimotor synchronization task there is a strong incentive to use sensory information for online error correction, but it is still done only partially by the individuals with autism, implying a broader influence of slow updating than was possible to conclude by the task of Lieder et al., 2019.

Third, the additional theoretical novelty of the current study is the direct test of the two opposing predictions of the “slow updating” hypothesis and the “increased volatility” hypothesis (Lawson et al., 2017). In the context of changing environments (Experiment 2), the “increased volatility” account predicts superior performance of individuals with autism, whereas the “slow updating” hypothesis predicts the opposite. Lieder et al. used constant frequency distributions in all studies, therefore they were unable to test performance under changes. In this study we introduce changing environments, which add another important incentive to use recent sensory information, since now inter-tap-intervals must also be updated, yet individuals with autism are slow also in this context.

We revised the Introduction so that these points are now better explained.

The authors conclude throughout their manuscript that the autism group demonstrate “poor sensorimotor synchronisation”. This seems a strange conclusion to me given that there was no difference in the mean asynchrony in Experiment 1. Greater variability does not seem to indicate poorer synchronisation. I.e., this result means that trials where they are slow to learn from error are being offset by other trials where they are especially accurate. Any idea why this mean difference is not observed? I am especially interested in how this also would link with the group difference in mean bias in the Lieder et al. (2019) because, given the common proposed underlying mechanism, I would have expected a mean group difference also in this study on the basis of those previous data.

In the finger tapping literature, mean asynchrony is believed to be determined mainly by low-level factors. This point is now better explained in the Introduction:

“...tapping is perceived as synchronous when individuals tap slightly ahead of each beat, hence the perceived synchrony is characterized by a small negative asynchrony (Aschersleben & Prinz, 1995; Repp 2005; Fig. 1a). The mean of this asynchrony is considered to reflect mainly low-level limitations, such as different processing latencies between auditory and somatosensory proprioceptive signals (Repp 2005; Repp & Su, 2013).”

Given this, we did not expect the mean to differ between groups.

Synchronization is measured by the ability to keep the asynchrony (interval between metronome beat and participant's tap) reliably across taps, or in other words, the consistency of the taps around this mean is what determines synchronization abilities. Importantly, we found that deviations are not corrected similarly in the different populations, despite the similar mean asynchrony values. Analysis of serial error correction shows that individuals with autism correct a smaller fraction of the error, leading to reduced ability to keep their taps perceptually aligned with the metronome beat. This point is shown in Figure 2, which plots online error correction between consecutive taps. A slope of 1 means no correction - an error of x ms in trial n predicts the same x ms error in trial $n+1$. All three populations have positive, smaller than 1, correlation slopes – meaning errors were only partially corrected. The slope that is closest to 1 is that of the group with autism. It means that errors produced by online noise will tend to keep and accumulate and interfere with synchronization to the metronome. Over a long block this might still result in comparable mean asynchrony values, but it indicates that synchronization abilities are markedly different between the groups. The concept of synchrony is now better explained in the Introduction.

Regarding the connection to Lieder et al. - We now explain more explicitly the relation between the tapping tasks and Bayesian theories in general and Lieder et al., specifically (second paragraph of the Discussion). Lieder et al., showed adequate pitch discrimination in autism, but reduced contraction bias by recent events. This means reduced tendency to perceive the pitch of the first tone in a pair as similar to that of previously presented tones. This reduction was with respect to recently presented tones, but not with respect to the broader mean pitch. We

termed it “slow update”. Here we asked whether slow updating can also explain “sluggish” motor performance. In line with the perceptual characteristics, the estimated mean interval is well perceived and maintained in autism (similar timekeeper noise in the model). Yet, the mean error correction – i.e. the magnitude of serial effect by the most recent event – is reduced. In the motor tasks – serial effect is the ability to quickly correct for the timing error in the previous tap. We find that the fraction of the previous-trial error which is corrected in the current trial is smaller in ASD than in both neurotypicals and individuals with dyslexia, which is in line with the results of Lieder et al.

Relatedly, another study demonstrates equivalent sensorimotor synchronisation in autism in a finger tapping task to metronome and should be incorporated. Edey et al. (2018; J Autism Developmental Disorders) found comparable absolute error between neurotypical and autistic groups when synchronising to an auditory metronome. However, they also found *lower* absolute error when synchronising to a visual metronome. How would the authors explain this difference between sensory modalities in interpreting their data? The Edey task included a different frequency metronome on each trial of four events, so if autistic individuals are slower to update models why do they show superior accuracy in this study?

We now discuss the results of the Edey et al. (2018) in detail in the Discussion (starting with “In the temporal domain, Edey et al. (2018) presented participants with 4 auditory (or visual) stimuli”). Briefly – Edey et al (in the auditory part of the paper) presented four auditory stimuli at a fixed interval. They asked participants to listen to the first two and press together with the third and fourth interval. Their mean asynchronies in these two presses did not differ from controls’. This result is consistent with our finding of the same mean asynchrony in controls and ASD. It would be interesting to assess the cross-trial variability around this interval, though sensitive assessments of serial effects, which are reduced in autism, require comparison across several tapping trials. It is interesting to mention that in a different temporal estimation task, Karaminis et al, (2016) studied the effect of previous trials’ intervals (context) on the estimation of the current interval and found reduced serial effects in children with autism.

Regarding ASDs’ better performance with a visual metronome, we can only speculate. Based on our hypothesis that ASDs’ better performance stems from underweighting of recent information, we propose that the recent prior in this specific experiment provided negative information as follows: The interval duration in the task varied between 300ms and 900ms on a trial-by-trial basis, hence neurotypical participants may have used this information to bias their responses towards the previous interval (one aspect of serial effects), while among ASDs this bias is reduced. When the order of intervals is chosen randomly, such a bias may hamper performance. This account is equivalent to the observations of Lieder et al., that performance in Bias- trials (trials in which recent info yields a bias that hampers performance) is better in ASD than in neurotypical groups. This account should apply to both the visual and auditory modalities and suggests that local statistics was somewhat different in these experiments, or that the baseline noise levels were different, requiring the neurotypical individuals to use serial dependencies less in the auditory task. Assessing whether this is indeed the case requires analysis of the raw data, to which we have no access. To conclude, slow update is sometimes

beneficial. Thus, if recent statistics provide a bad predictor of the coming stimuli (pitch, temporal intervals) – slow updating may improve performance since it implies less interference. Assessing whether this is the case requires analysis of the raw data. In daily situations recent statistics is typically a good predictor, but in lab conditions, where trials are often ordered randomly – it often is not.

Minor comments:

1. The study excludes many participants on the basis of certain scores on the WAIS, absolute reading scores, or ‘extreme’ asynchrony. Were these decisions pre-registered anywhere? It would seem more justifiable to do this according to standard deviations away from the mean.

We agree - overall 18% of our participants were excluded. This relatively large number resulted from the challenges of matching 3 groups with very different behavioral profiles, on age and Block Design. We had more participants with ASD whose Block Design scores were low (more than 2 standard deviations below scaled population mean), and hence had to be excluded. Still populations were not matched for Block Design. To keep as many ASD participants as possible with matched reasoning scores, we excluded participants with dyslexia and neurotypicals with very high Block Design scores. Regarding “extreme asynchrony” - it characterized only 3 / 133 participants. Keeping these participants in would lead to excluding most of their tapping data, which we considered a worse option. Our exclusion policy is now explained in more detail in the Methods section.

Importantly, this exclusion policy only weakened the results reported in the paper since neurotypical participants with higher Block Design scores tend to be better tappers (lower SD, better error correction) and individuals with ASD with lower Block Design scores tend to be poorer tappers.

2. When the authors describe the slow updating model in the introduction, they say that Bayesian updating may be “adequate”, yet the rate of updating of priors is “slow”. This sounds inadequate, rather than adequate, to me. I would therefore suggest the phrasing should be altered.

We modified the phrasing in the Introduction to - “An opposing account (“slow updating hypothesis”, Lieder et al., 2019) proposes that individuals with autism are able to estimate environmental statistics correctly, yet, the rate at which internal priors are updated is slow.”

3. In the regression with four time intervals included as regressors, I would assume that there would be a great deal of overlap in the explained variance by these four regressors – especially given the correlations presented earlier. Has this analysis therefore left out most variance from the analysis by adding all regressors together? Given the way this analysis is set up, it may make more sense to conduct a stepwise regression, where step 1 includes t-1, then

t-1 and t-2, then t-1, t-2, and t-3 etc. This way you ask whether additional variance is explained with the addition of each new step away from the present trial.

Thank-you for this suggestion. We have added a stepwise regression analysis (Results, paragraph beginning with “To understand the dynamics of phase correction we used an autoregressive model...” and section *Dynamics of phase correction using autoregressive modelling* of the supplementary material).

4. I would incorporate into the results (or introduction) that the metronome is auditory. This information seems crucial, given Edey 2018 (above), yet it was only mentioned for the first time in the methods at the end.

Done. Now made explicit in the first sentence of the Results section for both experiments.

REVIEWER COMMENTS

Reviewer #1 (Remarks to the Author):

I'm grateful to the authors for the great improvements and responses to my and the other reviewers' comments. The authors had to considerably modify (or tone down) the two main claims (discussed below) from the original manuscript, because they were insufficiently supported by the findings. This makes their paper much stronger logically and methodologically, but perhaps also less newsworthy. However, in my mind, this should never be grounds for rejection.

- With regard to the first claim, the problem was solved in a straightforward way: instead of motor plans the authors now talk more generally about slow updates of internal representations (could be perceptual, could be motor). That clearly better reflects the findings. Theoretically, it is what they also reported in their previous (Nature Neuroscience) study, albeit with a completely different paradigm/model.

- I'm also happy with the way they responded to the problems with the second claim: That is by openly admitting the limitations of their data/analysis, as well as by adding the data from the typical participants. The incomplete data in the autism group (as well as the lacking data in the dyslexia group) is very unfortunate. Especially because the dyslexia group is added to check for the specificity of the link between slow updating and social skills. So I'm skeptical about whether the correlation is meaningful, and implies what the authors make of it. There are three subscales in Austin (2005): communication/mind-reading, social skills, and details/patterns. This is probably better supported than the original AQ structure, but I would have been more at ease if the authors had used the Palmer et al study (DOI 10.1007/s10803-014-2289-1) as a basis, which had a much bigger sample (and similar but better-supported item groupings within factors). In Austin (2005), there are only six items in the communication subscale and one of them does not seem to fit there ("If there is an interruption, I can switch back to what I was doing very quickly"). Indeed this item does not appear in the mindreading factor in Palmer et al. In addition, it remains puzzling to me why the "social skills" factor does not correlate with slow updating. If mindreading is a capacity downstream from sensorimotor updating, social skills are so as well, so why no correlation? The reasoning given (in the introduction) for the hypothesized correlation with communication is flimsy or at least applies to social skills in general. In fact the same goes for the third (nonsocial) factor: Indeed, if slow updating is postulated as the core etiological factor in autism, then it should be related to autism (AQ score) severity in general, not just one subscale (this is what the authors claim at the very end: "Our results support the "slow-updating" account, which proposes that seemingly unrelated perceptual, motor and social characteristics of ASD can all be explained by slower integration of sensory information into representations that guide behavior."). After all, the nonsocial factor is centered on perceptual (pattern) and attentional (details) processes, so should be even *more* related to the processes investigated here. I'm sure that if the authors found such a general correlation with severity (total AQ score), they would have reported it. As it stands I'm worried about a false positive (knowing that it is based on few participants and few items). Of course, this does not invalidate the other findings, it just means that the questionnaire data are not unequivocal evidence for the general claim about slow updating as the underlying deficit in autism. In the current version, the authors already removed any claims about social functioning in the discussion, suggesting they are aware of the difficulties of such inferences given the current data. Reading this study carefully, I would agree it provides evidence for a 'slow updating' phenomenon in autism. I would not say it provides much evidence that "slow updating of internal representations contributes to both sensorimotor and communication difficulties in autism" (see abstract and sentence from end of discussion cited above). To talk about the latter, we would need more (complete) data than provided, and, crucially, a link with clinically reported sensorimotor issues in autism (as described by DSM and partly measured by the items of the AQ related to sensory sensitivity, attention to detail, need for structure/patterns, etc, *not* just the social aspects). Even just conceptually, it isn't made plausible how slow updating can lead to some of the classical symptoms and findings.

- All in all, I found the revision very well done, also with regard to the issues raised by the other reviewers. With exception of my point above, I find the authors do not overstate the findings anymore,

they clearly describe the (very sound) methodology, and so I consider this an important empirical contribution to the literature. I am not making judgments on broader significance of the findings. That will only be judgeable in retrospect.

Sander Van de Cruys

Reviewer #2 (Remarks to the Author):

The authors have done extensive work to address my concerns and replying my questions as well as answering the very pertinent remarks of the two other reviewers. I think the paper, its methods and overall clarity, are significantly improved as a result and will represent a valuable contribution to the field.

Reviewer #3 (Remarks to the Author):

I thank the authors for their thorough revision of the manuscript. Their responses to my concerns are logical and I believe the manuscript is now stronger. Just a couple of small remaining issues:

1. The authors have given a convincing account of the conceptual advance with respect to Lieder et al. 2019 and have revised their introduction accordingly. However this is only at the theoretical level and I could not find anywhere a description of the empirical findings in Lieder et al. 2019. Unless I missed something readers could take the earlier manuscript to simply contain a theoretical proposal. I propose that the authors add to their discussion briefly what was found in Lieder et al. 2019, to outline explicitly the empirical advance in the present dataset.

2. The authors also consider more extensively now why variability is taken to be especially relevant to understanding. However, the authors also say that “if difficulties involve central mechanisms” that a deficit should only be found in variability and not mean offset. The authors need to clarify this statement because I disagree with its current form. Central mechanisms influence mean offset too. This issue is discussed in Flach 2005 Human Movement Science. “The transition from synchronization to continuation tapping”. E.g., To give a real-world example, an orchestral piece will hang together with these mean offsets but only if every single player has an approximately comparable offset. Otherwise it would sound terrible. I remember Ruediger Flach telling me that this asynchrony is therefore smaller and sometimes absent in those with orchestral training, but this would have to be due to a role for central mechanisms rather than processing latencies disappearing. Apologies that I do not know a reference for this example, but the wider issue is discussed in Flach 2005.

3. Description of Edey et al 2018. The authors outline now the finding of equivalent mean offset in the auditory task of Edey et al. but my intrigue more related to the smaller mean offset in the visual task. I know any comparisons are only speculative but given the visual task is more comparable than the auditory I would suggest they use the discussion space to consider their speculative explanation of this *difference* between studies in the visual task rather than the null effects. I know it is speculative without analysis of the data but I think this is fine for the discussion. If the data are not online (quite plausible) I can send them over if they prefer to explore, but I think speculation at this stage is fine – although it would be interesting to examine empirically in the future.

Clare Press.

Reviewer #1 (Remarks to the Author):

I'm grateful to the authors for the great improvements and responses to my and the other reviewers' comments. The authors had to considerably modify (or tone down) the two main claims (discussed below) from the original manuscript, because they were insufficiently supported by the findings. This makes their paper much stronger logically and methodologically, but perhaps also less newsworthy. However, in my mind, this should never be grounds for rejection.

Thank you.

- With regard to the first claim, the problem was solved in a straightforward way: instead of motor plans the authors now talk more generally about slow updates of internal representations (could be perceptual, could be motor). That clearly better reflects the findings. Theoretically, it is what they also reported in their previous (Nature Neuroscience) study, albeit with a completely different paradigm/model.

Thank you.

- I'm also happy with the way they responded to the problems with the second claim: That is by openly admitting the limitations of their data/analysis, as well as by adding the data from the typical participants. The incomplete data in the autism group (as well as the lacking data in the dyslexia group) is very unfortunate. Especially because the dyslexia group is added to check for the specificity of the link between slow updating and social skills. So I'm skeptical about whether the correlation is meaningful, and implies what the authors make of it.

Thank you for raising this issue. We considered removing this section from the paper, but after careful consideration we believe it is meaningful and contributes to the manuscript (see below). However, we further toned down the claim by removing it from the abstract and the conclusion of the discussion. Additionally, we added the caveat that you raised regarding the specificity of the correlation. We now specifically state this in the relevant subsection of the Results and in a short paragraph of the Discussion (as detailed in the answer below).

We think the correlation is meaningful for the following reasons:

1. Thirty-seven neurotypicals were added. Their update rate was correlated with their self-report of mindreading/communication ability. The 19 participants with ASD also showed the expected correlation. Their similar dependence is

- informative. Additionally, the mindreading factor we use separates very nicely (most separating factor in our population) between the two populations.
2. Regarding the dyslexia data – while it would be nicer to have the dyslexia data too – since we do not find (and obviously do not claim) that the relation between slow update and mindreading difficulties is unique to autism, we feel this is less crucial. What is unique to autism is their particularly slow update rate – which we clearly show in the paper with respect to both neurotypical and dyslexic populations.
 3. Assessing the mindreading/communication subscale is based on recent literature, which shows that both prosocial tendency (Cirelli et al., 2018), and mindreading/communication abilities are promoted by sensorimotor synchronization (Baimel et al., 2015; Tarr et al., 2014 and more). Baimel et al. (2015) proposes that moving in synchrony tunes our minds for reasoning about *other* minds. Baimel et al. 2018 reports increased self-reported tendencies and abilities for considering others' mental states following synchronous activity. This literature is now explicitly referenced in the Introduction (paragraph beginning with "Our choice of task was also motivated..."), and in the specific subsection of the Results "Update rate is correlated with communication and mindreading skills" (first sentence).

These studies predict that prosocial and mindreading tendencies would both be correlated with synchrony. We find a strong correlation with mindreading (cognitive), but not with the social skill (emotion) subscale. We are not sure why, and now state both in this subsection of the Results ("We hypothesized that slower update rate (...) would correspond to poorer social and/or communication skills."), and in a short paragraph of the Discussion (beginning with: "Our observation of synchronization difficulties...").

There are three subscales in Austin (2005): communication/mind-reading, social skills, and details/patterns. This is probably better supported than the original AQ structure, but I would have been more at ease if the authors had used the Palmer et al study (DOI 10.1007/s10803-014-2289-1) as a basis, which had a much bigger sample (and similar but better-supported item groupings within factors). In Austin (2005), there are only six items in the communication subscale and one of them does not seem to fit there ("If there is an interruption, I can switch back to what I was doing very quickly"). Indeed this item does not appear in the mindreading factor in Palmer et al. In addition, it remains puzzling to me why the "social skills" factor does not correlate with slow updating.

First, we agree with you regarding the social aspect. We are not sure why this is the case, and explicitly state it in the revised manuscript (as described above). Informally, though, we experienced a clear dissociation between joy from socializing, and mindreading ability: about half of our participants with autism enjoyed social situations, such as talking in length to the experimenter, though they showed no sensitivity regarding the experimenter's interest, and the conversation was mainly a monologue.

Regarding the number of items in the mentalizing/mindreading factor: both Palmer's and Austin's have 6 items in this factor. Austin's 6-item factor very nicely separates between the neurotypical and ASD populations (Cliff's delta for mindreading is 0.63 (Figure 8a) and is 0.49 and 0.45 for subscales 1 and 2, respectively). Palmer's mentalizing subscale is a slightly poorer separator (Cliff's delta 0.6).

Regarding the content of these factors: it is interesting that they share only 2 items, perhaps most importantly the following item - "I find it difficult to work out people's intentions". But, Austin has items we consider relevant (though we agree about the 6th being not relevant), that do not appear in Palmer's: "Other people frequently tell me that what I've said is impolite, even though I think it is polite."; "People often tell me that I keep going on and on about the same thing." and "I am often the last to understand the point of a joke." We agree that none of the factors is perfect. But for the above reasons, we consider Austin's mindreading factor better and given that it is much more cited (even normalized per year), we think it is adequate to base our analysis on this factor. By the way, we are now administering another experiment, and Austin's communication factor nicely separates between the populations.

If mindreading is a capacity downstream from sensorimotor updating, social skills are so as well, so why no correlation? The reasoning given (in the introduction) for the hypothesized correlation with communication is flimsy or at least applies to social skills in general. In fact the same goes for the third (nonsocial) factor: Indeed, if slow updating is postulated as the core etiological factor in autism, then it should be related to autism (AQ score) severity in general, not just one subscale (this is what the authors claim at the very end: "Our results support the "slow-updating" account, which proposes that seemingly unrelated perceptual, motor and social characteristics of ASD can all be explained by slower integration of sensory information into representations that guide behavior.").

This sentence is now modified to: "Our results support the "slow-updating" account, which proposes that slow update of internal representations is a core deficit of autism, contributing to both perceptual and motor difficulties."

After all, the nonsocial factor is centered on perceptual (pattern) and attentional (details) processes, so should be even *more* related to the processes investigated here. I'm sure that if the authors found such a general correlation with severity (total AQ score), they would have reported it. As it stands I'm worried about a false positive (knowing that it is based on few participants and few items).

Indeed, AQ as a whole was not correlated, but this is expected given that AQ50 is not a unidimensional measure of autistic traits (Lundqvist & Lindner, 2017). Many studies have shown that the different subscales are not correlated, or even anti-correlated (Shuster et al., 2014). While social and communication abilities are sometimes grouped together, repetitive and restricted behaviors have consistently been shown to be a distinct trait,

which is often anti-correlated with the social aspects of the AQ scale (this is also true for the Palmer 2015 subscales, which show a mild, yet significant, negative correlation between the Detail Orientation subscale and the other two subscales, see table 3 in their paper).

Given this dissociation between the AQ subscales, and the specific literature implicating social abilities to be connected with synchronization, we believe a specific correlation with social\communication skills is more expected than a correlation with the entire AQ.

Lastly, despite including few items, Austin's communication\mindreading subscale is a reliable separator between the autistic and neurotypical populations (Cliff's delta is 0.63, see Figure 8a). The entire AQ50 scale separates between the groups only slightly better (Cliff's delta is 0.69, see first paragraph of this Results subsection). Regarding the number of participants, note that the newly added neurotypicals (37 additional participants) showed the same pattern, bringing the total number of participants in the joint correlation to n=56.

Of course, this does not invalidate the other findings, it just means that the questionnaire data are not unequivocal evidence for the general claim about slow updating as the underlying deficit in autism. In the current version, the authors already removed any claims about social functioning in the discussion, suggesting they are aware of the difficulties of such inferences given the current data.

Right, we are aware of the difficulties, and now state them explicitly in the manuscript (as specified above).

Reading this study carefully, I would agree it provides evidence for a 'slow updating' phenomenon in autism. I would not say it provides much evidence that "slow updating of internal representations contributes to both sensorimotor and communication difficulties in autism" (see abstract and sentence from end of discussion cited above). To talk about the latter, we would need more (complete) data than provided, and, crucially, a link with clinically reported sensorimotor issues in autism (as described by DSM and partly measured by the items of the AQ related to sensory sensitivity, attention to detail, need for structure/patterns, etc, *not* just the social aspects). Even just conceptually, it isn't made plausible how slow updating can lead to some of the classical symptoms and findings.

We expanded the conceptual link in the Introduction, Results and Discussion sections:

Introduction: "Our choice of task was also motivated by previous studies showing that synchronization is functionally related to theory of mind (Baimel et al., 2015; Baimel et al., 2018) and to social behavior (Cirelli, 2018) in non-clinical populations. The rationale proposed for these observations is that synchronized actions promote a predictive mechanism trained to anticipate other's actions and wants (Keller et al., 2014; Novembre

et al., 2019). In line with these observations, other studies report impaired synchronization in autism (both in social and non-social contexts – Fitzpatrick et al, 2017; McNaughton & Redcay, 2020; Morimoto et al., 2018).”

Results: “Since previous literature suggests that synchronization is associated with social skills (e.g. Baimel et al., 2018; Cirelli, 2018), we asked whether slower updating is correlated with these skills among our participants in the neurotypical and autism group...We hypothesized that slower update rate... would correspond to poorer social and/or communication skills.”

Discussion: “Our observation of synchronization difficulties in a non-social context indicate that poor synchronization is not a unique outcome of a lack of social interest (Jaswal & Akhtar, 2019). Rather, reduced synchronization may reduce the interest in other people's state of mind, though causality is likely to operate in both directions. We found a correlation between our measure of update rate and mindreading skills, in both neurotypicals and people with ASD, yet we did not find a significant correlation with social joy. There is also other evidence for distinct processes underlying the neurocognitive vs affective influences on social skills (Mogan et al., 2017). Therefore it is possible that update rate taps onto one mechanism, but not all. Further studies, which include direct clinical measures, are needed to clarify the functional relations.”

Given the a-priori justification, the strength of the correlation and stating the caveats explicitly, we feel we are on "safe grounds".

- All in all, I found the revision very well done, also with regard to the issues raised by the other reviewers. With exception of my point above, I find the authors do not overstate the findings anymore, they clearly describe the (very sound) methodology, and so I consider this an important empirical contribution to the literature. I am not making judgments on broader significance of the findings. That will only be judgeable in retrospect.

Sander Van de Cruys

Dear Sander, thank-you for your thoughtful comment. We believe we have addressed all your concerns regarding the correlation section.

Reviewer #2 (Remarks to the Author):

The authors have done extensive work to address my concerns and replying my questions as well as answering the very pertinent remarks of the two other reviewers. I think the paper, its methods and overall clarity, are significantly improved as a result and will represent a valuable contribution to the field.

Thank you.

Reviewer #3 (Remarks to the Author):

I thank the authors for their thorough revision of the manuscript. Their responses to my concerns are logical and I believe the manuscript is now stronger. Just a couple of small remaining issues:

1. The authors have given a convincing account of the conceptual advance with respect to Lieder et al. 2019 and have revised their introduction accordingly. However this is only at the theoretical level and I could not find anywhere a description of the empirical findings in Lieder et al. 2019. Unless I missed something readers could take the earlier manuscript to simply contain a theoretical proposal. I propose that the authors add to their discussion briefly what was found in Lieder et al. 2019, to outline explicitly the empirical advance in the present dataset.

We added a description of the empirical findings of Lieder et al. 2019 in the second paragraph of the introduction and further clarified the novelty the empirical advance in the current study also in the third paragraph of the discussion (paragraph starting with "This observation suggests an underweighting of recent sensory information into a form that can be used to guide behavior").

2. The authors also consider more extensively now why variability is taken to be especially relevant to understanding. However, the authors also say that "if difficulties involve central mechanisms" that a deficit should only be found in variability and not mean offset. The authors need to clarify this statement because I disagree with its current form. Central mechanisms influence mean offset too. This issue is discussed in Flach 2005 Human Movement Science. "The transition from synchronization to continuation tapping". E.g., To give a real-world example, an orchestral piece will hang together with these mean offsets but only if every single player has an approximately comparable offset. Otherwise it would sound terrible. I remember Ruediger Flach telling me that this asynchrony is therefore smaller and sometimes absent in those with orchestral training, but this would have to be due to a role for central mechanisms rather than processing latencies disappearing.

Apologies that I do not know a reference for this example, but the wider issue is discussed in Flach 2005.

Thank-you for this clarification. We agree that the previous phrasing was over simplistic, and modified the introduction, to better clarify the sources of the mean asynchrony, as follows:

"The mean of this asynchrony is influenced by many factors, including the magnitude of the movement, the type of feedback, and the sound of the metronome (Aschersleben & Prinz, 1995; Aschersleben, 2002; Repp, 2005; Flach, 2005; Danielsen et al., 2019). The

relative contribution of peripheral and central sources is not known; hence we had no prediction for group differences. Second, the timing of tapping around this mean is variable. Though tapping variability is also affected by both central (e.g. intelligence, Ullén et al., 2008; Madison et al., 2009), and peripheral factors, such as motor noise, the contribution of peripheral factors is smaller (Jacoby et al., 2015). Importantly, the underlying components of tapping variability have been extensively modelled, in a manner which allows a separation between the different contributions (Wing & Kristofferson, 1973; Mates, 1994; Vorberg and Wing 1996; Vorberg and Schulze 2002). Therefore, we focused our investigations on this aspect of sensorimotor synchronization.”

As you mentioned, many studies, including in our lab (Weiss et al., 2014) have shown absent or reduced negative mean asynchrony in expert musicians. Interestingly, musicians show the same effects when tapping alone, in a non-orchestral context, which is what we found. However, learning too involves many different components, central and peripheral.

3. Description of Edey et al 2018. The authors outline now the finding of equivalent mean offset in the auditory task of Edey et al. but my intrigue more related to the smaller mean offset in the visual task. I know any comparisons are only speculative but given the visual task is more comparable than the auditory I would suggest they use the discussion space to consider their speculative explanation of this *difference* between studies in the visual task rather than the null effects. I know it is speculative without analysis of the data but I think this is fine for the discussion. If the data are not online (quite plausible) I can send them over if they prefer to explore, but I think speculation at this stage is fine – although it would be interesting to examine empirically in the future.

We speculate that different noise levels in auditory and visual tasks may underlie differences in serial dependency between the tasks. In the case of Edey et al., visual noise seems to be larger than auditory noise, as suggested by poorer performance in the visual modality. Larger noise is expected to lead to stronger serial dependencies, which may hamper performance (perhaps) to a larger degree in the neurotypical population.

We extended the Discussion of Edey et al., 2018 to include this explanation:

“A difference in serial dependency profiles between the groups may also underlie the higher accuracy of the autism group in the visual condition observed by Edey et al. It has been shown in several contexts that visual sensorimotor synchronization is noisier than auditory sensorimotor synchronization (Hove et al., 2013; Repp, 2005; Repp & Su, 2013), which may lead participants, particularly neurotypicals to increase the magnitude of serial dependency (Cicchini et al., 2018), and perhaps consequently hamper their performance (e.g. Lieder et al., 2019).”

Clare Press.

REVIEWER COMMENTS

Reviewer #1 (Remarks to the Author):

I thank the authors for another round of great responses and edits in the paper, and have no further concerns. Their treatment now has the necessary nuance in the intro and discussion, as well as sound methods and interesting findings, which together makes for an great contribution to the literature.

Best wishes,
Sander

Reviewer #3 (Remarks to the Author):

The authors have addressed all my remaining concerns and I therefore recommend publication in its present form. I believe this manuscript will make an interesting contribution to the literature.

Clare Press.